# Machine learning identifies scale-free properties in disordered materials

Sunkyu Yu[1,2], Xianji Piao [1] & Namkyoo Park [1✉]

The vast amount of design freedom in disordered systems expands the parameter space for signal processing. However, this large degree of freedom has hindered the deterministic design of disordered systems for target functionalities. Here, we employ a machine learning approach for predicting and designing wave-matter interactions in disordered structures, thereby identifying scale-free properties for waves. To abstract and map the features of wave behaviors and disordered structures, we develop disorder-to-localization and localization-to-disorder convolutional neural networks, each of which enables the instantaneous prediction of wave localization in disordered structures and the instantaneous generation of disordered structures from given localizations. We demonstrate that the structural properties of the network architectures lead to the identification of scale-free disordered structures having heavy-tailed distributions, thus achieving multiple orders of magnitude improvement in robustness to accidental defects. Our results verify the critical role of neural network structures in determining machine-learning-generated real-space structures and their defect immunity.

[1] Photonic Systems Laboratory, Department of Electrical and Computer Engineering, Seoul National University, Seoul 08826, Korea. [2] Intelligent Wave Systems Laboratory, Department of Electrical and Computer Engineering, Seoul National University, Seoul 08826, Korea. ✉email: nkpark@snu.ac.kr

Disordered systems cover all regimes of structural phases, including periodic, quasiperiodic, and correlated or uncorrelated disordered structures, each of which has its carefully tailored strength and pattern of disorder. The classification of disorder according to microscopic structural information has thus attracted great attention in various fields, such as many-body systems[1], network science[2], and wave–matter interactions[3]. In wave physics, rich degrees of freedom in disordered systems enable exotic wave phenomena distinct from those of periodic or quasiperiodic systems, including strong[4] or weak[5] localizations, broadband responses in wave coupling[6] or absorption[7], and topological transitions with disorder-induced conductivity[8]. In particular, localization phenomena have received an extensive amount of attention as the origin of material phase transitions[9] and as the toolkit for energy confinement[3,10,11] that enables multimode lasing[12] and nanoscale sensing[13].

Traditional approaches for exploring disordered structures and their related wave behaviors have employed mapping between disordered structures and wave properties through different types of mathematical microstructural descriptors[1], such as $n$-point probability, percolation, or cluster functions. Each descriptor unveils a specific aspect of structural patterns, which enables the classification of disordered structures according to their correlations and topologies and reveals the origin of distinct wave behaviors in each class of disorder. By including the descriptors in the cost function for the optimization process, numerous inverse design methods have also been developed for generating disordered structures from target wave properties: stochastic[1,14], genetic[15], or topological[16] optimizations. However, traditional approaches are still challenging owing to the large design freedom inherited from disordered structures; thus, these approaches require very time-consuming and problem-specific processes to extract microstructural information at each stage of iterative and case-by-case design procedures. Until now, most works have focused on lower orders of microstructural descriptors (for example, two- or three-point probability functions) due to the significant complexity in calculating and interpreting higher-order descriptors[1]. However, even such simple descriptors have stimulated intriguing concepts and dynamics for disordered structures, such as hyperuniformity[17–19] for disordered bandgap materials[20].

To substitute the time-consuming and problem-specific process of calculating analytical microstructural descriptors while making full use of microstructural information, we can envisage the use of multiple-layer neural network (NN) models as data-driven descriptors to identify the relationship between disordered structures and wave behaviors. This deep-learning-based framework[21,22], one of the powerful machine-learning (ML) tools, has proven successful for abstracting the features of data sets in pattern recognition, decision making, and language translation[23,24] when carefully preprocessed data can be used. Because of its applicability to general-purpose data formats, deep learning has recently been extended to handle a number of physics problems[25,26], such as classifications of crystals[27] or topological order[28], phase transitions and order parameters[29–31], optical device designs[32–37], and image reconstructions[38]. When we consider the vast amount of design freedom in disordered systems, deep learning will compose a powerful toolkit for resolving complexities in wave behaviors inside disordered structures, as shown in the inference of phases of matter using eigenfunctions[26,31].

Here, we employ deep convolutional neural networks (CNNs)[39] to identify the physical relationships between disordered structures and wave localization. The prediction of localization properties in disordered structures and the generation of necessary structures for target localizations are achieved with disorder-to-localization (D2L) and localization-to-disorder (L2D) CNNs, respectively, by transforming disordered structures to multicolor images. Using dropout[40] or L2 regularization[22] techniques to avoid overfitting, the CNNs implemented with Google TensorFlow[41] are successfully trained with the expanded training data set of collective and individual lattice deformations, even drawing an extrapolatory inference for the untrained regimes of disorder. Most importantly, our CNN-based generative model identifies disordered structures with scale invariance following the power law. The heavy-tailed distributions in these scale-free structures lead to an increase of two to four orders of magnitude in robustness to unexpected structural errors when compared to conventional disordered structures having normal distributions. We show that the ML-generated scale-free material with hub atoms inherits the properties of robustness to accidental attacks (or defects) and relative fragility to targeted attacks (or modulations)[42], in contrast to the democratic robustness of conventional normal-random disordered structures. The proposed approach can be applied to discover unexplored regimes of disorder in general wave systems and paves the way towards the design of materials by manipulating the ML architecture or the training process of NN structures.

## Results

**Imaging disorder and localization.** We consider disordered structures obtained from the random deformation of a finite-size, two-dimensional (2D) square lattice of identical atoms (from Fig. 1a, b). Each atomic site of the lattice can describe a quantum-mechanical wavefunction of an atom, a phononic resonance of a metamaterial, or a propagating mode of an optical waveguide. The standard tight-binding Hamiltonian of an $N$-atomic system governed by the eigenvalue equation $\mathbf{H}\Psi_m = E_m\Psi_m$ ($m = 0, 1, \ldots, N - 1$) is

$$\mathbf{H} = \sum_i \varepsilon \hat{a}_i^\dagger \hat{a}_i + \sum_{i,j} (t_{ij}\hat{a}_i^\dagger \hat{a}_j + \text{h.c.}), \qquad (1)$$

where $\varepsilon$ is the on-site energy, $\hat{a}_i^\dagger$ (or $\hat{a}_i$) is the creation (or annihilation) operator in the $i$th lattice site, $t_{ij}$ is the random hopping integral between the $i$th and $j$th lattice sites ($1 \le i, j \le N$), and h.c. denotes the Hermitian conjugate. The disordered pattern is described by $t_{ij}$, which is determined by the spatial distance $d_{ij}$ between the $i$th and $j$th lattice sites. For generality, we consider all orders of hopping between lattice sites by defining the near-field hopping condition $t_{ij} = t_0\exp(-\alpha d_{ij})$, where the coefficients $t_0$ and $\alpha$ are determined by an individual atomic Wannier function[43]. The distance $d_{ij}$ is adjusted by the perturbation on the position of each atom site (see Eq. (5) in "Methods" section).

To develop D2L and L2D CNNs for the inference of wave–matter interactions, we devise a multicolor image representation of a disordered structure to be used as the CNN input. In this scenario, a 2D random displacement of an atomic site is projected along $x$ and $y$ spatial axes ($\Delta x$ and $\Delta y$ in Fig. 1c), and the resulting two ($x$ and $y$) projected layers from the entire disordered structure are assigned as two-color images for CNNs (Fig. 1d, e). This projection can be directly extended into a 3D disordered structure, which leads to the sets of three-color images with a tensor form.

The localization property of the proposed structure is quantified by the normalized mode area[44] $w_m$, which is defined by the inverse of the inverse participation ratio (IPR) as

$$w_m = \frac{1}{N} \frac{\left[\sum_{s=1}^N (\psi_m^s)^2\right]^2}{\sum_{s=1}^N (\psi_m^s)^4}, \qquad (2)$$

where $\psi_m^s$ denotes the $s$th component of the eigenstate $\Psi_m$ ($s = 1, 2, \ldots, N$). The operation of the CNNs will then be the inference of the relationships between two-color images (disordered

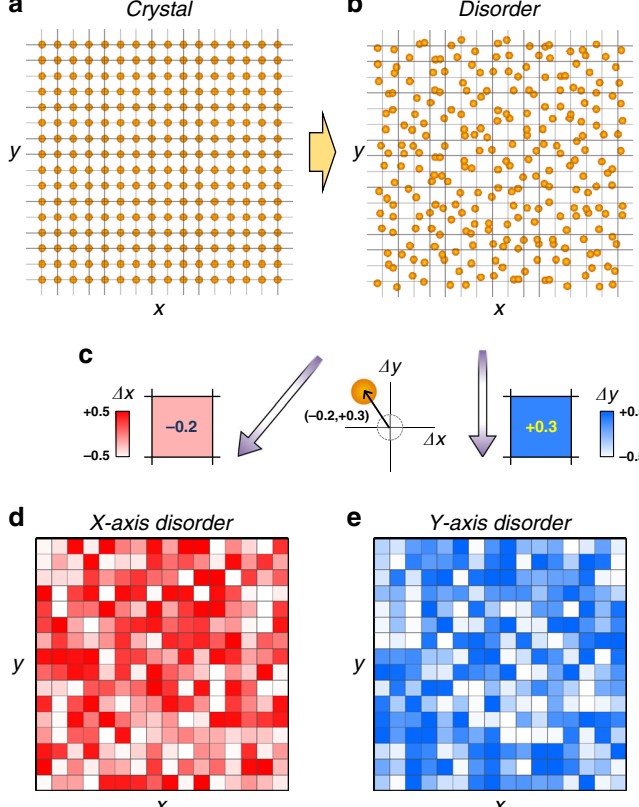

**Fig. 1 Multicolor image representation of disordered structures. a** A two-dimensional (2D) square lattice crystal and **b** its deformation that generates a disordered structure. **c** The projections of the 2D displacement of each atomic site along the $x$ and $y$ axes ($\Delta x$ and $\Delta y$), which define the pixel values of the $x$ axis and $y$ axis color images, respectively. **d**, **e** The resulting two-color images obtained from the disordered structure in **b**. **d** The red-to-white image for the $x$-axis projection $\Delta x$ and **e** the blue-to-white image for the $y$-axis projection $\Delta y$.

structures) and a 1D array (mode area). The 1D mode area array is reshaped into a single-color 2D image when it is used as the input to the L2D CNN, as discussed later.

**Disorder-to-Localization CNN.** Figure 2a shows the network structure of the D2L CNN. For the two-color image input, the CNN is composed of 3 cascaded convolution-pooling stages and the fully connected (FC) layer in front of the $N$-neuron output layer for the 1D array of $w_m$ (see "Methods" section for network parameters). Each convolution-pooling stage is a series of the convolution (Conv) layer with $3 \times 3$ filters to extract a feature map and the max-pooling layer to reduce the feature map size[21,22,39]. Because each mode has different degrees of localization, it is necessary to fairly estimate the regression error for a wide range of $w_m$ values. We thus employ the mean absolute percentage error (MAPE) as the cost function, which has been widely applied to regression and machine learning for forecasting models[45,46]. The MAPE cost function for the D2L CNN is expressed as

$$L_{\mathrm{D2L}} = \sum_{m=0}^{N-1} \frac{\left| w_m^{\mathrm{True}} - w_m^{\mathrm{ML}} \right|}{w_m^{\mathrm{True}}}, \qquad (3)$$

where $w_m^{\mathrm{ML}}$ is the D2L-CNN-calculated mode area and $w_m^{\mathrm{True}}$ is the ground-truth mode area calculated by the Hamiltonian **H** in Eq. (1).

The CNN is trained with the training data set of randomly deformed lattices and their localization properties. The expanded training sets of $2 \times 10^4$ realizations are obtained by introducing both collective and individual deformations of atomic sites to improve the inference ability of the CNN (see "Methods" section, Supplementary Note 1, and Supplementary Fig. 1 for details of the deformation process). The validation accuracy of the CNN defined by $1 - L_{\mathrm{D2L}}$ is monitored with the validation data set of $1 \times 10^4$ realizations during the training. After training with the error backpropagation method[47], we calculate the test accuracy $1 - L_{\mathrm{D2L}}$ of the trained CNN with the test data set of $1 \times 10^4$ realizations (see "Methods" section, Supplementary Note 2, and Supplementary Figs. 2–4 for the extended discussion of the training process, such as avoiding overfitting and selecting the cost function). To monitor overfitting during and after the training, different random seeds for the deformation have been used in the training, validation, and test data sets.

Through the training process, we successfully trained D2L CNN to predict disorder-induced localization. Figure 2b–e shows the ground-truth and ML prediction of the mode areas $w_m$ from given disordered structures: nearly crystallized (or weak disorder) (Fig. 2b, d) and nearly random (or strong disorder) (Fig. 2c, e) structures. We also compare the ground-truth (Fig. 2f) and ML-predicted (Fig. 2g) localization for a wide range of localization values of the test data set ($1 \times 10^4$ realizations). Figure 2f, g is obtained by plotting $w_m$ of each realization as a function of the mode number $m$ and coloring each point according to the average mode area $w_{\mathrm{avg}} = \sum_{m=1}^{N} w_m / N$ of each realization. We note that the ground-truth and ML-predicted localization shows excellent agreement for different values of $w_{\mathrm{avg}}$, achieving the test accuracy $1 - L_{\mathrm{D2L}} \sim 94.80\%$. The trained D2L CNN enables an almost instantaneous prediction of localization properties for each mode from a given disordered structure without solving the eigenvalue problem of the Hamiltonian **H** in Eq. (1).

**Localization-to-disorder CNN.** As demonstrated in a classic question[48] of "Can one hear the shape of a drum?" and its answer[49], the relationship between a wave property (such as the localization or eigenspectrum) and material (or structural) platforms is non-unique, allowing multiple possible structures for a given wave property. This one-to-many relationship between a wave property and matter has made it difficult to achieve a stable inverse design of material from a given wave property because the existence of many solutions (matter) for an input (wave property) prohibits the stable convergence of the optimization for a cost function. In the inverse design of material using the ML method, several different approaches have been proposed to resolve this non-uniqueness problem: training of the input through a trained NN[32], training of the inverse NN from a trained forward NN[33,34], reinforcement learning[35], and iterative design of multiple NNs for each family of material structures with a given scattering property[36]. Considering the large design freedom in disordered structures, we employ the second approach[33,34]: training of the inverse L2D CNN using the pre-trained forward D2L CNN.

Figure 3a shows the network structure of the L2D CNN. The L2D CNN has the same network configuration as the D2L CNN (three convolution-pooling stages and the FC layer), except for the input and output layer (see "Methods" section for network parameters). The results of the L2D CNN from the $2N$ output neurons are reshaped to the two-color images that represent the spatial profile of the ML-generated disordered structure. To guarantee the physical reality of the obtained solution, we utilize the trained D2L CNN with the fixed weight and bias parameters, which instantaneously predicts the localization in ML-generated disordered structures. The connection of the L2D CNN with the

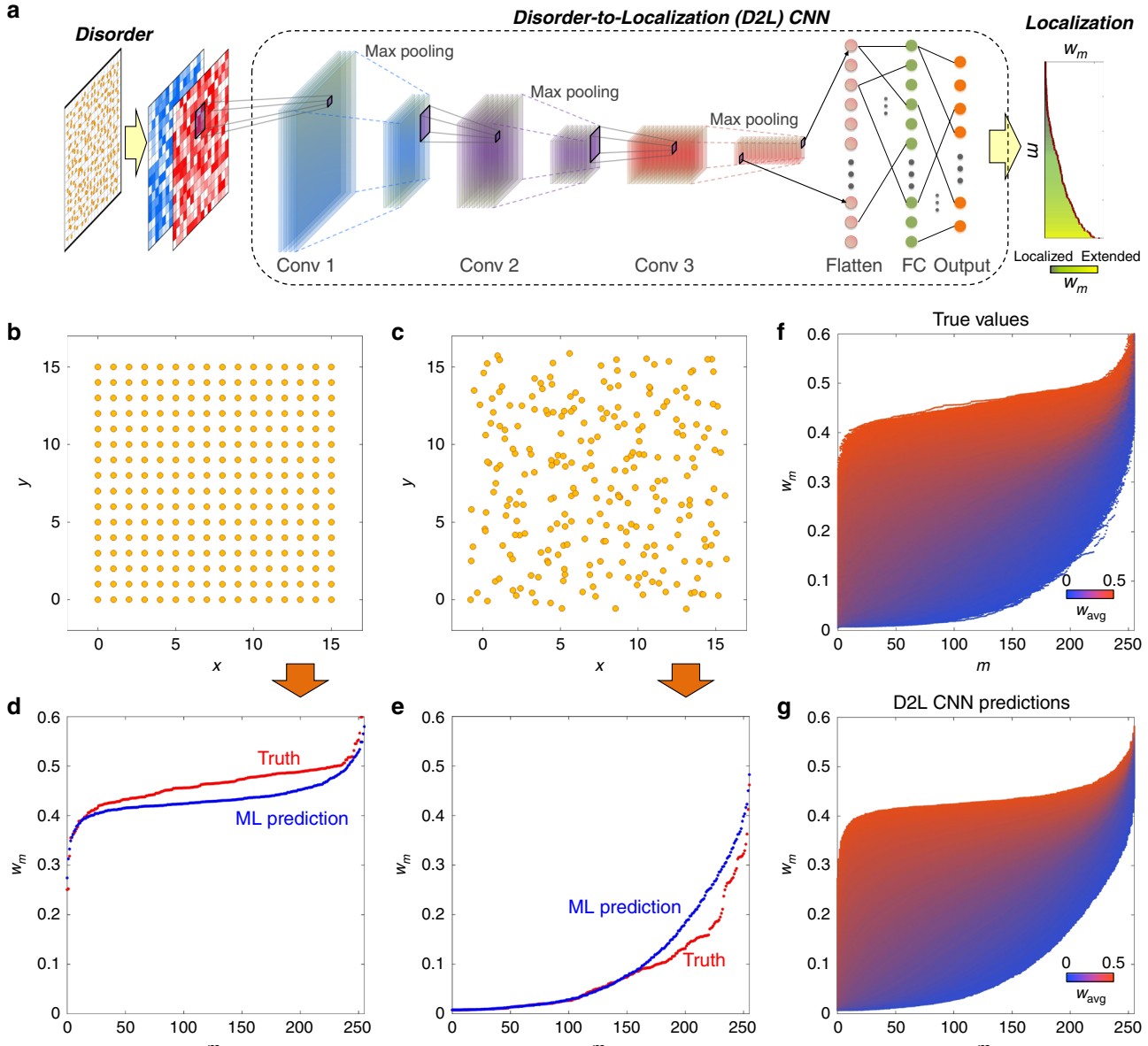

**Fig. 2 Disorder-to-localization convolutional neural network for predicting localization. a** The network structure of the disorder-to-localization (D2L) convolutional neural network (CNN). The details of the network parameters are shown in the "Methods" section. **b–e** The prediction of localization properties $w_m$ for **b**, **d** weakly disordered, and **c**, **e** strongly disordered structures. **f**, **g** Comparison of localizations between **f** the ground truth and **g** the D2L CNN prediction for a broad range of average mode area $w_{avg}$ of the test data set ($1 \times 10^4$ realizations). The period of the 16 × 16 unperturbed square lattice (256 atoms) is set to 1 and the hopping parameters are $t_0 = 3.14 \times 10^{-2}$ and $\alpha = 1.1454$ throughout the manuscript. Although the on-site energy does not affect mode areas, we set the on-site energy as $\varepsilon = 1$ for energy spectra in the later discussion (Supplementary Note 6). Mode numbers $m$ are sorted according to localization values in all examples throughout the manuscript.

trained D2L CNN constructs the localization-to-disorder-to-localization (L2D2L) network (Fig. 3b), which effectively operates as the autoencoder for localization data. The MAPE cost function of the L2D2L CNN is defined as

$$L_{\text{L2D2L}} = \sum_{m=0}^{N-1} \frac{\left| w_m^{\text{Target}} - w_m^{\text{ML}} \right|}{w_m^{\text{Target}}}, \quad (4)$$

where $w_m^{\text{ML}}$ is the mode area calculated by the L2D2L CNN and $w_m^{\text{Target}}$ is the target mode area. The training of the entire L2D2L CNN (i.e., the partial training of the L2D CNN part) then allows the generation of disordered structures for the target wave localization (see "Methods" section, Supplementary Note 2, and Supplementary Fig. 2 for the training process, including the

comparison between the validation and training accuracies). Training, validation, and test data sets are again prepared with different random seeds. We note that although the training data set for the L2D2L CNN consists of localization data obtained from the tight-binding Hamiltonian in Eq. (1), the microstructural information used for the target localization data is not applied to the training of the L2D2L CNN.

The trained L2D CNN achieves a high test accuracy of $1 - L_{\text{L2D2L}} \sim 94.21\%$. We compare the target localizations (Fig. 3c) to the ML-predicted localizations obtained through the L2D2L CNN (Fig. 3d) and the Hamiltonian-calculated true values of the disordered structures generated by the L2D CNN (Fig. 3e), using the same data plotting format with those in Fig. 2f, g. Despite the good agreement between the target and true values (~79.10%

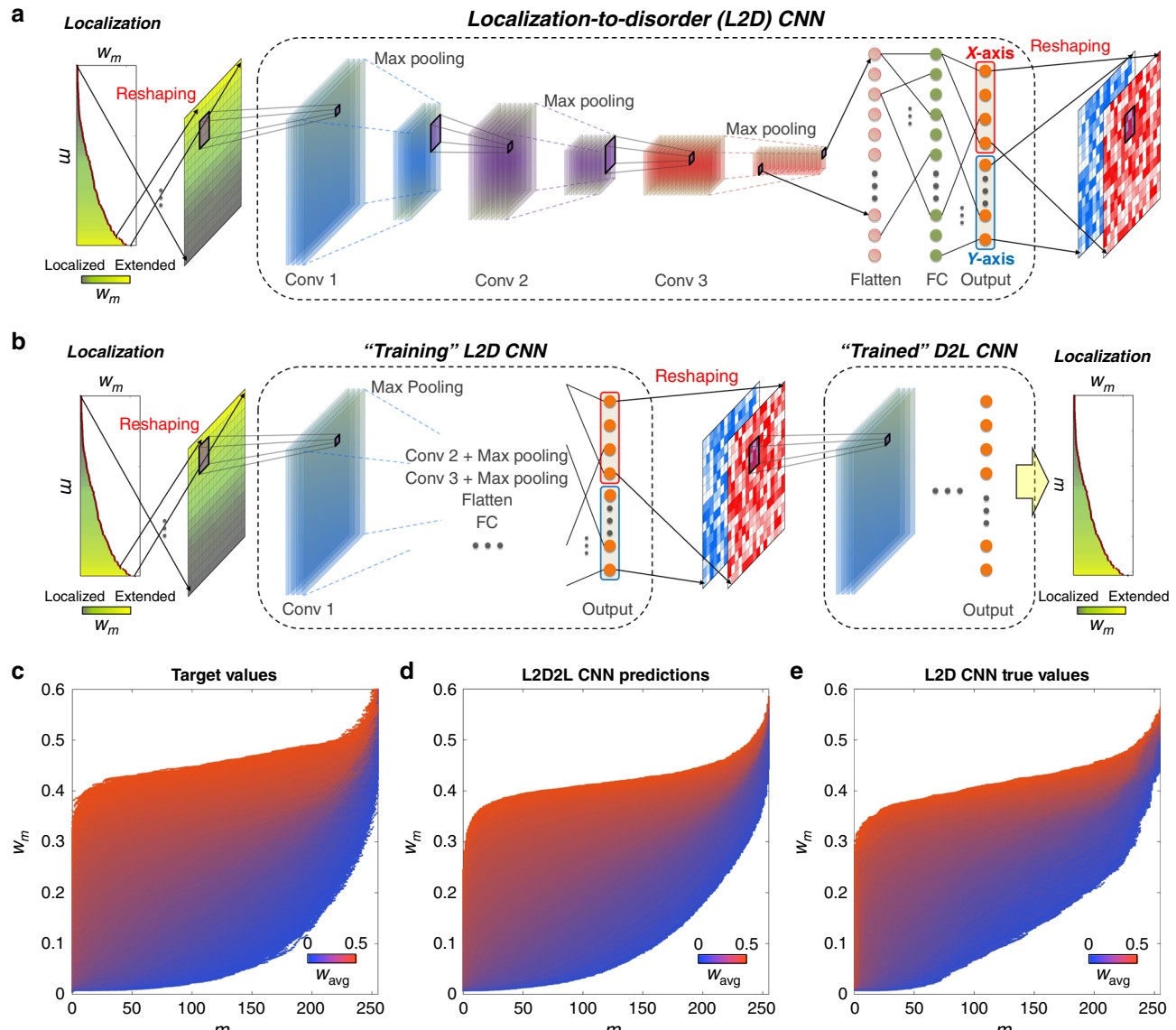

**Fig. 3 Localization-to-disorder convolutional neural network for generating disordered structures. a** The network structure of the localization-to-disorder (L2D) convolutional neural network (CNN). **b** The network structure of the localization-to-disorder-to-localization (L2D2L) CNN for training the L2D CNN with the pre-trained disorder-to-localization (D2L) CNN. The details of the network parameters are shown in the "Methods" section. **c–e** Comparisons of localizations between **c** the target values, **d** the machine-learning-predicted values from the L2D2L CNN, and **e** the Hamiltonian-calculated true values with the disordered structures generated by the L2D CNN for a broad range of average mode area $w_{avg}$ of the test data set ($1 \times 10^4$ realizations).

between Fig. 3c, e), a non-negligible discrepancy exists near the strong localization regime with large deformations of atomic sites. We note that this test accuracy degradation originates from the emergence of large deformations in the L2D-CNN-generated structure, which easily exceeds the maximum deformation value inside the training data sets for the D2L CNN. Therefore, the test accuracy of the L2D CNN is restricted by the limit of the extrapolation: the inference of the untrained regime of localization. The current good extrapolation could be further improved by expanding the range and type of training data sets and the number of hidden layers. However, we emphasize that large deformations themselves unveil a very intriguing but little recognized property in ML inverse designs[32–37]: the effect of the NN structure on the ML-generated real-space structure, which enables the identification of scale-free properties for waves, as discussed in the later sections.

**Scale invariance in ML-generated microstructures**. Due to the one-to-many relationship between a wave property and matter, the obtained ML-generated disordered structure corresponds to only one realization among numerous possible options for the target wave property. To examine the property of this ML identification, in Fig. 4a–f, we compare the ML-generated structure with a seed structure having very similar localization properties. For the regimes of weak (Fig. 4a–c) and strong (Fig. 4d–f) disorder, we use initial seed structures (Fig. 4a, d) to obtain the target localization (red curves in Fig. 4c, f). By employing this target localization as an input of the trained L2D CNN, we achieve the corresponding ML-generated structures (Fig. 4b, e), which represent localization properties that are very similar to those of seed structures (black curves in Fig. 4c, f). However, surprisingly, the ML-generated structures consist of lattice deformations that are evidently different from the original

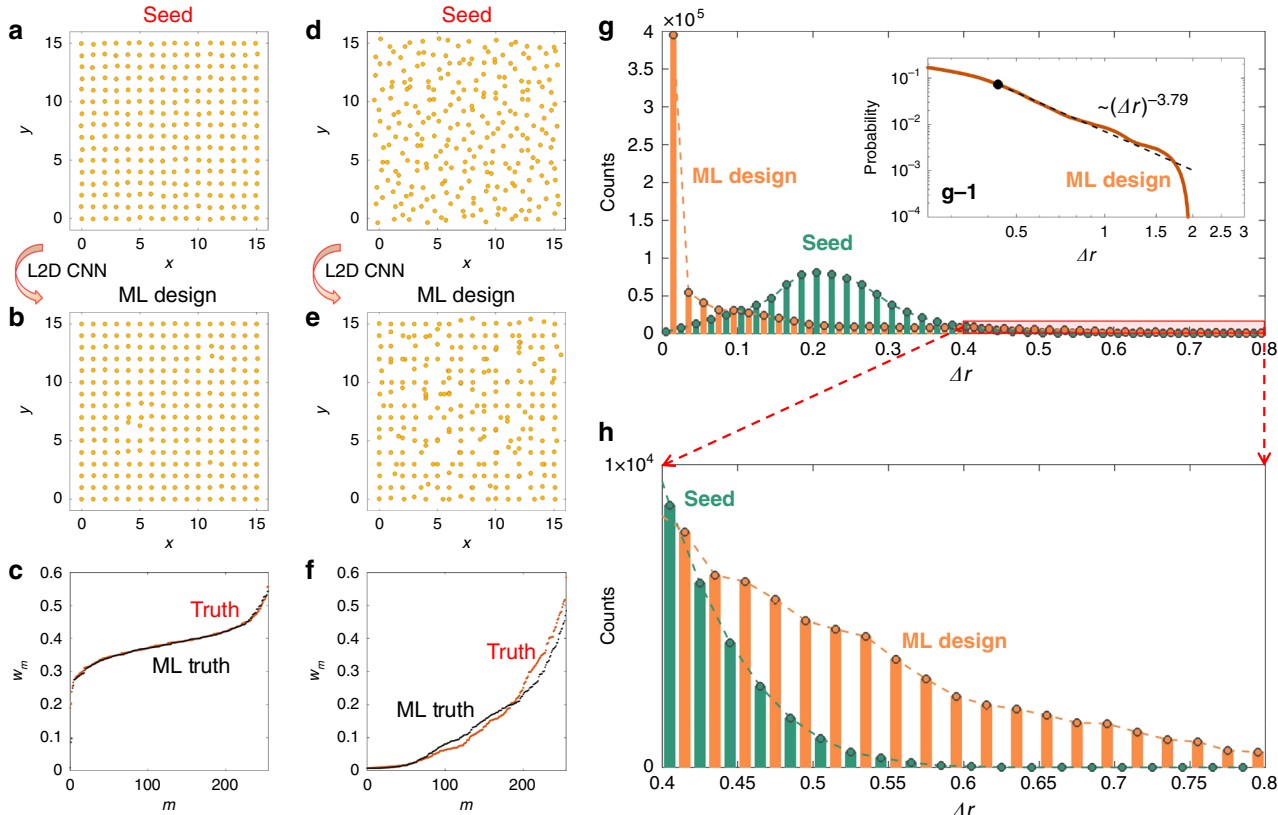

**Fig. 4 Scale-invariant disordered structures generated by the localization-to-disorder convolutional neural network. a–f** Comparison between seed and machine-learning- (ML-) generated structures for **a–c** weak and **d–f** strong disorder. **a**, **d** Seed structures that provide the target localizations for the localization-to-disorder (L2D) convolutional neural network (CNN). **b**, **e** ML-generated disordered structures obtained from the L2D CNN. **c**, **f** Localization of seed ("Truth", red dotted lines) and ML-generated ("ML Truth", black dotted lines) structures, obtained from the tight-binding Hamiltonian in Eq. (1). **g** Statistical distributions of the strength of the lattice deformation $\Delta r$ in the seed (green) and ML-generated (orange) structures for 3200 realizations satisfying $0.20 \leq w_{avg} \leq 0.30$ in the ML design. The first inset g-1 shows the log–log plot of **g** for the ML design, illustrating the power-law distribution. The orange line (composed of discretized points) represents the complementary cumulative distribution function (CDF) obtained from the data set in **g**. The black dashed line represents the best fit to the data using the method in refs. [50,51], showing the power-law fitting of $(\Delta r)^{-3.79}$. The black dot represents the lower bound $\Delta r_{min} = 0.432$ to the power-law behavior. **h** The extended plot of the range $0.4 \leq \Delta r \leq 0.8$ in **g** demonstrating the heavy-tailed distribution of the ML design.

deformations in the seed structures. This result originates from the training process of the L2D CNN, which is achieved from the training of the L2D2L CNN using only localization data (Fig. 3a) without the data of seed microstructures. The identification of the microstructure from the target localization can then have many possible options and is determined by the network structure of the L2D CNN, as discussed later.

For a deeper understanding of the differences between seed and ML-generated structures, we analyze the microstructural statistics of disordered structures by counting the distributions of the atomic site deformation $\Delta r_i = [(\Delta r_i^x)^2 + (\Delta r_i^y)^2]^{1/2}$, where $\Delta r_i^x$ and $\Delta r_i^y$ are the displacements of the $i$th atom along the $x$ and $y$ axes, respectively ($1 \leq i \leq N$; see Eq. (5) in "Methods" section for seed structures, whereas $\Delta r_i$ of ML-generated structures is obtained from the L2D CNN). Figure 4g shows the microstructural statistics of the seed and ML-generated structures for 3200 realizations where the ML-generated structures have an average mode area $w_{avg}$ in the range of $0.20 \leq w_{avg} \leq 0.30$. We note that the seed and ML-generated structures show apparently differentiated statistics. First, the microstructural statistics of the seed structures follows a normal distribution due to the definition of Eq. (5) in Methods. However, the analysis based on the maximum-likelihood fitting method with goodness-of-fit tests[50,51] shows that the ML-generated class follows power-law statistics $(\Delta r)^{-\alpha}$ (inset (g-1) of Fig. 4g) and possesses a

heavy-tail distribution (Fig. 4h). To guarantee the reliability of the power-law fitting result, in Supplementary Note 3 and Supplementary Fig. 5, we analyze the power-law exponent $\alpha$ and the lower bound of the heavy tail $\Delta r_{min}$ for a different number of realizations. The result shows that the unique statistical distribution of ML-generated structures is maintained for a small number of realizations, from roughly $10^1$ (2560 atoms) to $10^2$ (25,600 atoms) realizations, and even a single realization also provides a similar value of $\alpha$ and $\Delta r_{min}$.

The result in Fig. 4g, h demonstrates that ML-generated disordered structures are composed of scale-invariant deformation without the characteristic perturbation strength of $\Delta r$. This finding is in sharp contrast to the characteristic $\Delta r$ of seed disordered structures, which is defined as the statistical center of their normal distribution. We note that the scale invariance of ML-generated disordered structures is universally observed for varying degrees of localization (Supplementary Note 4 and Supplementary Figs. 6 and 7), which strongly implies that the identification of scale-invariant disordered structures originates from the properties of the L2D CNN, not from the observed wave–matter interactions. In Supplementary Note 5 and Supplementary Fig. 8, we also study the fitting with other heavy-tailed distributions[2,50], such as a power-law distribution with an exponential cutoff and a log-normal distribution, again

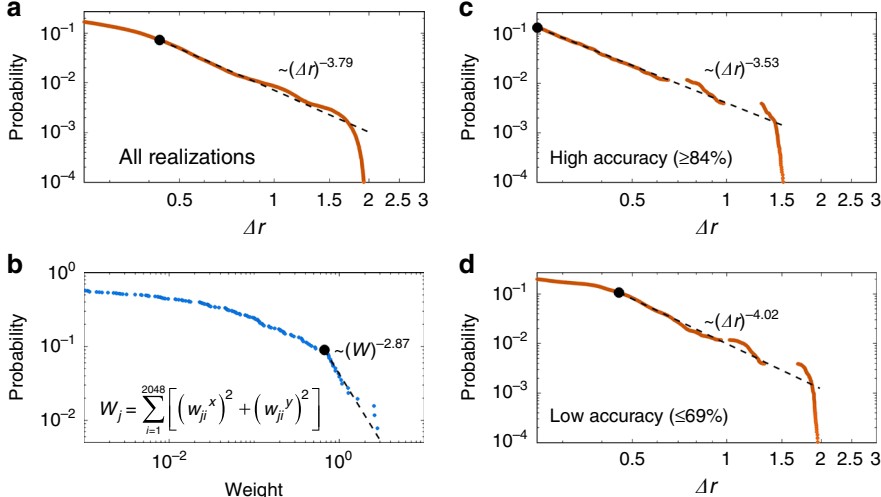

**Fig. 5 Scale invariance in machine-learning-generated disordered structures and neural network structures. a** Power-law fitting of the statistical distribution of $\Delta r$ in machine-learning-generated disordered structures, which is the same figure with Fig. 4g-1 and is shown for comparison. **b** Power-law fitting of the statistical distribution of the weight strength parameter $W_j$. **c, d** Power-law fitting results of the realizations for **c** high ($\geq 84\%$) and **d** low ($\leq 69\%$) test accuracies. All of the fitting results are based on the same method using in Fig. 4g-1.

confirming the reliability of the power-law fitting and the observed scale-free invariance.

Furthermore, the seed and ML-generated structures show very similar localization properties and distinct energy spectra (see Supplementary Note 6 and Supplementary Fig. 9 for energy spectra). Therefore, the L2D CNN enables the independent and systematic handling of a part of wave quantities: here, the conservation of localization with an altered energy spectrum through the transformation of microstructural statistics from normal-random to scale-invariant distributions. On the other side, among various possible realizations of disordered structures for a given wave property (here, localization) due to the one-to-many relationship between a wave and matter, the L2D CNN successfully selects one particular realization, which notably has the scale invariance in the structural profile.

Because the values of the output neurons in the L2D CNN determines the lattice deformation in ML-generated structures, the scale invariance in the deformation is strongly related to the NN structure (weight and bias distributions) of the L2D CNN. To examine this conjecture, in Fig. 5a, b, we analyze the relationship between the microstructural statistics of ML-generated structures and the network structure of the L2D CNN, including an ablation study. Among numerous weight and bias parameters (roughly $1.5 \times 10^7$ parameters each in the D2L and L2D CNNs), the most critical parameters are the weights from the FC layer (2048 neurons) to the output layer (512 neurons) in the L2D CNN, which are described by $2048 \times 512$ matrix. Although the weights and bias in hidden layers should also affect the output layer neurons indirectly, we expect that this indirect effect is less significant than the direct effect from the FC-output weights.

For $w_{ji}^x$ and $w_{ji}^y$, which denote the weights from the $i$th FC neuron to the $j$th $x$ axis and $y$ axis output neurons, respectively ($1 \leq i \leq 2048$ and $1 \leq j \leq 256$ in our design), we define the strength of the weights to the $j$th output neuron (or the $j$th atom in an ML-generated disordered structure) as $W_j = \Sigma_i[(w_{ji}^x)^2 + (w_{ji}^y)^2]$. Figure 5b shows the CDF of $W_j$, which represents a very similar statistical distribution with $\Delta r$ in terms of its inflection point (Fig. 5a) and also possesses the heavy-tailed distribution. The comparison between Fig. 5a, b provides clear-cut evidence of the effect of the NN structure on ML-generated materials. This finding becomes more evident by examining different ML architectures which lead to different weights and bias

distributions. In Supplementary Note 7 and Supplementary Figs. 10–12, we conduct an ablation study by investigating another D2L and L2D CNN each with a single-pooling stage, which enables the control of the $W_j$ distribution and the following alteration of ML-generated structures. We note that the heavy-tailed distribution is also maintained in this single-pooling-layer design.

To guarantee the generality of the observed scale-free properties, we also examine the effect of the test accuracy on the scale invariance (Fig. 5c, d). Among 3200 realizations in the example in Fig. 4, we select the sets of ML-generated structures having high ($\geq 84\%$, 194 realizations, Fig. 5c) and low ($\leq 69\%$, 191 realizations, Fig. 5d) test accuracies. We note that both cases possess very similar statistical distributions with the power-law fitting result. This result again confirms that the scale invariance originates from the statistical distribution of the ML architecture, not from the mismatch between the ML result and theoretical truth.

**Scale-free materials with heavy tails and hub atoms**. The scale invariance in microstructural statistics (Figs. 4 and 5) imposes intriguing characteristics on ML-generated disordered structures: "scale-free" properties on waves. Scale-free properties, which represent the power-law probabilistic distribution with heavy-tailed statistics, have been one of the most influential concepts in network science[2,52], data science[50,51], and random matrix theory[53,54]. In addition to its ubiquitous nature in biological, social, and technological systems[2], the most important impact of scale-free property is the emergence of core nodes, also known as "hubs", which possess a very large number of links or interactions, thereby governing signal transport inside the system[2,42,52]. The existence of hub nodes strongly correlates with the robustness of scale-free systems: fault-tolerant behaviors, especially superior robustness to accidental attacks and relative fragility to targeted attacks[2,42,55], which can also be extended to other heavy-tailed distributions (Supplementary Note 5) without the perfect scale-free (or power-law) features.

Although the scale-free nature is well-defined in the infinite-size limit[2,42,52], similar to the condition of ergodicity in random heterogeneous materials[1], the power-law microstructural statistics of our systems with the heavy-tailed distribution leads to well-defined hub behaviors and the following robustness of wave

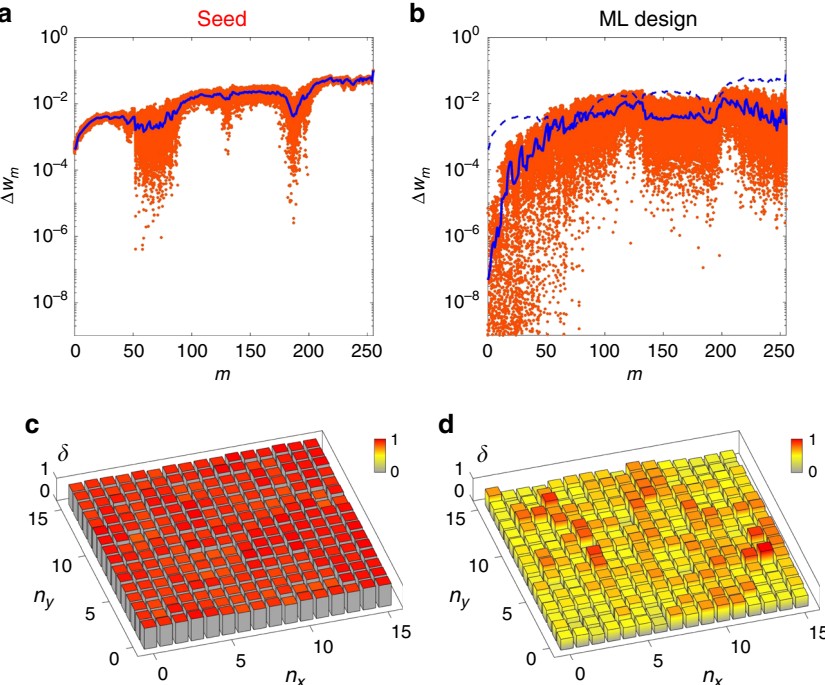

**Fig. 6 Robustness and sensitivity of scale-free machine-learning-generated disordered structures. a, b** Comparison of the robustness in **a** seed and **b** machine-learning- (ML-) generated disordered structures in terms of the perturbation in the mode area from the attack (or error) to a specific atom. Each red point denotes the perturbation of the $m$th mode area $\Delta w_m$ by imposing the attack to a specific atom. Blue solid lines represent the average perturbation. The blue dashed line in **b** is the average perturbation of the seed structure shown in **a** for comparison. **c, d** Normalized errors $\delta$ for attacking each atom in the **c** seed and **d** ML-generated disordered structures. Larger $\delta$ denotes a more sensitive response of wave localization to the attack. $n_x$ and $n_y$ denote the $x$ and $y$ indices in the unperturbed square lattice, respectively ($0 \leq n_x$, $n_y \leq 15$ for 256 atoms).

properties. To investigate the robustness of our wave systems, we exert the "attack" (material imperfection, system error, or modulation) on each atom of disordered structures to adjust their localization properties. The attack is defined by the position perturbation of each atom as $r_i^x = r_i^{x0} + \rho_a \cos[u_i(0, 2\pi)]$ and $r_i^y = r_i^{y0} + \rho_a \sin[u_i(0, 2\pi)]$, where $r_i^{x,y}$ (or $r_i^{x0,y0}$) are the $x$ and $y$ perturbed (or original) positions of the $i$th atom in a disordered structure, $\rho_a$ is the perturbation strength, and $u_i(p, q)$ is the random value for the $i$th atom from the uniform random distribution between $p$ and $q$.

Figure 6a, b shows the degree of robustness in two disordered structures with different microstructural statistics in terms of the perturbation of localization $\Delta w_m$. The attack is applied to each atom of normal-random seed ($w_{avg} = 0.145$) and scale-free ML-generated ($w_{avg} = 0.140$) disordered structures, which have similar localization properties (~84.05% test accuracy). Remarkably, compared with the seed structure, the scale-free disordered structure shows a reduction of two to four orders of magnitude in the perturbation of mode areas $\Delta w_m$, especially in highly localized modes (small $m$). This result demonstrates that the scale-free ML-generated disorder provides more robust localization properties than the normal-random seed disorder, following fault-tolerant behaviors in general scale-free systems[2,42,55].

In Fig. 6c, d, we also demonstrate the existence of hub atoms, which is the origin of the robustness of scale-free systems[2,42]. To detect hub atoms in disordered structures, we define the normalized error $\delta$ that measures the average perturbation of the mode area $\Delta w_m$ obtained by attacking a specific atom. First, the apparent democratic response of $\delta$, which represents the nearly equal perturbation of $\Delta w_m$ regardless of the perturbed atom position, is observed in the normal-random seed structure (Fig. 6c), following the signal behavior in Erdős–Rényi random systems[2]. In contrast, our ML-generated scale-free disordered

structure is no longer democratic; some hub atoms derive more sensitive responses (larger $\delta$) to the perturbation (Fig. 6d), following the signal behavior in Barabási-Albert scale-free systems[2,42]. This result successfully demonstrates the scale-free nature of our ML-generated disorder: highly robust localization to accidental perturbations and relatively fragile localization to targeted perturbations on hub atomic sites. Notably, because ML-generated disorder partly exhibits imperfect scale-free, but heavy-tailed distributions, the relationships between the scale-freeness, heavy-tailed distributions, and the defect robustness and modulation sensitivity will require further study.

## Discussion

Because the ML-generated lattice deformation is strongly related to the weights of the output neurons in the L2D CNN, the apparent stochastic difference between normal-random seed structures and scale-free L2D CNN outputs raises an interesting open question; the training process of deep NNs could inherently possess the scale-free property. Recently, in random matrix theory, it was demonstrated that the correlations in the weight matrices of well-trained deep NNs can be fit to a power-law with the heavy-tailed distribution[53,54]. This theory enables the successful analogy between NN structures and ML-generated real-space wave structures in our result: the identification of the "heavy-tailed perturbation distribution" of atomic sites using the "heavy-tailed weight distribution" of CNN neurons. While these complex systems in software and real-space emphasize the role of the "heavy tail" in the statistical distribution, the optimization process of the CNNs in this viewpoint corresponds to the evolutionary process of realizing general scale-free systems[2,42,52]. We also note that exploring ML architectures to control scale-free properties or even realize non-scale-free distributions will inspire exciting future research in material science and wave physics. For

the inverse design of disordered systems and the following statistical analysis of ML-generated materials in terms of scale-free properties, the applications of reinforcement learning, unsupervised learning, or well-trained NNs such as U-net[56,57] would be an excellent topic for study. Notably, the utilization of an attention mechanism and the transformer architecture[58] would also be helpful to model the relationships between atomic information in disordered structures or wave localization, as similar to an attention score to model the influence each word has on another in natural language processing.

In terms of interpreting tight-binding lattices as graph networks[59–61], the change in lattice deformations through the ML method (from Fig. 4a, d to Fig. 4b, e) can be explained as the change of isoperimetric parameters[62]: the relative size of graph vertex subsets to the size of their boundary. Our result then corresponds to the control of isoperimetric parameters while preserving a wave property (here, localization), which enables the independent control of other wave properties (here, error robustness). The further study on graph properties of ML-generated structures is thus necessary to clarify the relationship between physical systems, their graph representations, and the ML-based design.

In terms of the previous studies[63–65] on disordered structures with power-law correlation distributions, the power-law exponent $\alpha$ is closely related to localization lengths (Supplementary Note 4) and the emergence of an Anderson-like metal-insulator transition. Because we employed the 2D seed structures with an uncorrelated disorder, which eventually lead to Anderson localization according to the scaling theory of localization[66], the allowed range of the power-law exponent should be restricted due to the similar degrees of localization. The finding of ML-generated structures with more tunable $\alpha$ is then necessary to extend the regime of disorder achieved by the ML approach. This goal would be enabled by utilizing seed structures that break the traditional assumptions in the scaling theory[67], using inhomogeneity, anisotropy, and inelastic scattering.

In conclusion, we demonstrated that the ML approach can identify disordered materials with the target localization, which also have scale-free properties for waves. Instead of calculating microstructural descriptors for analyzing disordered structures, we proposed a CNN-based modeling approach for wave–matter interactions, by using convolution processes in CNNs to abstract and map the relationship between localization and disordered structures. With successful training results for the ML prediction and generation of wave–matter interactions, we showed that ML-generated disordered structures possess scale invariance with power-law microstructural statistics, which is the result of the structural properties of the ML architecture. We demonstrated that the ML-generated disordered structures can operate as scale-free materials for waves with excellent robustness in terms of wave behaviors and hub dynamics. Scale-free materials, or, more broadly, the materials with heavy-tailed distributions discovered by the ML method will stimulate a new design strategy for general wave devices in disordered structures, such as lasing[12], energy storage[68], and complete bandgap materials[20]. Scale invariance can significantly improve the performance of these wave devices by achieving robustness to accidental errors (such as unwanted defects in fabrications or measurements) and the fragility to targeted errors (such as the intended system modulation for active devices). Along with the ML generation of scale-free structures with target wave properties, our results will motivate further research on controlling CNN training or selecting different CNN architectures, which will enable the generation of wave structures analogous to various types of complex systems, such as small-world, modular, or self-similar systems. The obtained scale-free wave material will also offer new insight into other scale-free-type material structures, such as Lévy glasses with superdiffusion[69,70]: the microstructural realization of a random walk having step lengths with a power-law distribution.

## Methods

**Neural network structures and training hyperparameters of D2L and L2D CNNs.** For $N = 16 \times 16$ atomic lattices, the D2L CNN accepts two $16 \times 16$ images as the input (a disordered structure), whereas the L2D CNN accepts a single $16 \times 16$ image as the input (a reshaped mode area). For both D2L and L2D CNNs, the numbers of filters (or the thicknesses) of the convolution layers are set to 256, 512, and 1024 in the first, second, and third layers, respectively. We use zero padding to maintain the spatial dimensions of feature maps during the convolution processes[22,39]. The max-pooling layer leads to the down-sampling of feature maps by extracting the maximum value of each patch with a stride of 2 pixels[21,22]. The result of three cascaded convolution-pooling states is reshaped (or flattened) to a 1D array and is connected to the FC layer, which has 2048 neurons. The FC layer is connected to the $N$-atomic output layer in the D2L CNN for the mode area $w_m$ and is connected to the $2N$-atomic output layer in the L2D CNN for two-color images that describe a disordered structure.

To avoid a vanishing gradient problem during training, we use the rectified linear unit (ReLU) activation for each layer of CNNs. We utilize the Adam optimization function[47] with an exponential decay in the learning rate for stable convergence and employ a mini-batch of size 10 for efficient learning. To avoid overfitting, we apply the dropout method[40] in the D2L CNN by randomly keeping 50% of neurons in the FC layer during training and apply the L2 regularization[22] in the L2D CNN (TensorFlow scale parameter: 0.05) to suppress excessively large values of weights. The learning processes of the D2L and L2D CNNs are shown in Supplementary Note 2. All ML computations were performed on a single desktop computer with two NVIDIA GeForce RTX 2080 Ti GPUs.

**Deformation of lattices for data sets.** To train the CNNs, avoiding overfitting to a certain type of disordered structures, the carefully preprocessed training data set has to cover a wide range of the relationship between disordered structures and localization from large to small values of $w_{\mathrm{avg}} = \sum_{m=1}^{N} w_m/N$. For this purpose, we assign the collective and individual deformations of atomic sites as

$$\Delta r_i^x = \rho \cos(u_i(0, 2\pi)) + u_i(-\sigma, +\sigma),$$
$$\Delta r_i^y = \rho \sin(u_i(0, 2\pi)) + u_i(-\sigma, +\sigma), \tag{5}$$

where $\Delta r_i^x$ and $\Delta r_i^y$ denote the displacements of the $i$th atom along the $x$ and $y$ axes ($1 \leq i \leq N$), respectively; $u_i(p, q)$ is the random value for the $i$th atom from the uniform random distribution between $p$ and $q$; $\rho$ is the amplitude of the collective displacement of all atoms, and $\sigma$ is the amplitude of the individual displacement of each atom. The strengths of the collective and individual deformations are randomly assigned for each realization of the data set, as $\rho = \rho_{\max}u(0, 1)$ and $\sigma = \sigma_{\max}u(0, 1)$, where $u(a, b)$ is the random value assigned to each realization from the uniform random distribution between $a$ and $b$. We set $\rho_{\max} = 0.6$ and $\sigma_{\max} = 0.6$ for all examples in this manuscript. The comparison between collective and individual deformations through different values of $\rho_{\max}$ and $\sigma_{\max}$ are shown in Supplementary Note 1.

## Data availability
The data that support the plots and other findings of this study are available from the corresponding author upon request.

## Code availability
All code developed in this work will be made available from the corresponding author upon request.

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

## Acknowledgements

We acknowledge financial support from the National Research Foundation of Korea (NRF) through the Global Frontier Program (S.Y., X.P., N.P.: 2014M3A6B3063708), the Basic Science Research Program (S.Y.: 2016R1A6A3A04009723), and the Korea Research Fellowship Program (X.P., N.P.: 2016H1D3A1938069), all funded by the Korean government.

## Author contributions

S.Y. and N.P. conceived the idea presented in the manuscript. S.Y. and X.P. developed the theory and ML codes using Google TensorFlow. N.P. encouraged S.Y. and X.P. to investigate disordered systems for waves using ML and network theory while supervising the findings of this work. All authors discussed the results and contributed to the final manuscript.

## Competing interests

The authors declare no competing interests.
