## [Peer Review File · Nature Communications]

Reviewers' comments:

Reviewer #1 (Remarks to the Author):

This paper presents a neural network design to learn and convert disorder in a 2d lattice to electron localization modes (waves) and vice-versa. The methodology is mostly sound, but some architectural design choices, as well as choices in training procedure need clarification for me. Also, I am a bit skeptical about their interpretation of the observation about the fat-tailed disorder distribution in the parts where ML doesn't match the theory. If I find their clarifications satisfactory, I would recommend this for

Comments:

The idea on training ML with random functions and asking it to produce target functions is great and should be used more in material design and other branches of physics. I like the idea of the paper, though the scale-free aspect is a strange observation. I am surprised and don't know if I find enough insight in the paper as to why ML chooses these solutions. If the authors manage to convince me they understand why this happened and what is going on I may recommend this for publication. Right now, there are quite a few seemingly arbitrary choices in the architectural design as well as this surprising observation which may well be just an artifact of the architecture. My detailed comments below.

1. My guess is that the scale-free part has to do with the peculiar architectural design chosen by the authors. For example, the 3 levels of max-pooling coarse-grains the disorder by a factor of 8. This exponential decrease in size for each maxpool may allow "fat-tailed" distribution of disorder have similar effects to a normally distributed one. But do the two types of disorder really produce the same physics, or is it just that this ML design is not capable of telling them apart?

2. Also, I am a bit opinionated about the choice of architectural design, as I outline below, but as a first work and proof of concept I think their design is ok. In general I am positive about this paper and think it presents a nice way of using synthetic data with random initial conditions to learn a nonlinear map between two phenomena, in this case disorder and localization wave pattern, and then use the learned mapping to produce a desired wave pattern. I have personal experience with this approach in a different context and it seems to work very well.

3. The way the wave-to-matter CNN is trained is also interesting. Usually, one encounters such a design in generative Adversarial networks (GAN). But in GAN, one trains the generator (wave2matter) and the discriminator (matter2wave) at the same time, because otherwise the trained discriminator would be so good that there would be no hope for the generator to converge. So, I am a little surprised that it works so well with a fixed, pre-trained matter2wave. I suspect that is because of the continuous nature of the w_m used in the loss function. Any comments?

4. Are "accuracies" test accuracies? It should be made clear that they are not simply training accuracy.

5. Fig 3. a) what is the intuition behind the design? At first sight, you are building an auto-encoder for the localization. But you also want it to be a decoder which converts localization to disorder, hence the reshaping, etc. What I don't understand is why you first make the localization wave spatial, then flatten it to a fully connected network and then reshape again to get the disorder image. I guess you want the FC layers to learn the non-linear wave-to-matter transformations, right?

6. Architecture, personal opinion:

Matter-to-Wave CNN:

Both the mode area array and the disorder have size = N . In the Matter-to-Wave CNN, why do you not try to convert w_m to 16×16 and use convolutional output layers, as you would do with an

auto-encoder or the U-net architecture?

U-Net, in particular has been observed to do very well with converting two functions over a lattice to each other.

2. You have a 16×16 input. After 3 layers with maxpooling, you have at most a $2 \times 2 \times (\text{num filters})$ output, so most spatial info is extremely coarse-grained. Then, instead of a traditional upsampling CNN and 16×16 w_m output, you are using FC layers to get a vector w_m . Can you motivate this? Do you try any other architecture for your decoder?

Reviewer #2 (Remarks to the Author):

The authors have shown that convolutional neural network (CNN) can predict the inverse participation ratio (IPR) of wavefunctions from the distribution of atoms (Matter-Wave CNN), as well as the distribution of atoms from the IPR (Wave-Matter CNN). The obtained distributions of atoms show scale-free properties, which is demonstrated to be robust against perturbations, namely, the fluctuations of positions of atoms. The results are interesting, but already more than three years have passed since the numerous machine learning techniques was applied to condensed matter physics. I therefore have reservation to recommend publication of this article in Nature Communications since the impact of this research is no longer matches the high publication criterion. The paper may be more suitable in publication in Scientific Reports, but before I recommend publication, I have several comments/questions that might improve the paper and its readability.

1) In equation (2), w_m is written that it is defined by the IPR. This sentence is a bit confusing, since w_m is the inverse of IPR, not IPR itself. The authors should write explicitly that w_m is the inverse of IPR.

2) In Fig. 2, the on-site energy ϵ is set to 1. Do we need this? The energy ϵ is just the shift of whole energy spectrum, and the localization properties should be independent of ϵ .

3) In the section of "scale-free wave networks with hub atoms," the position perturbation of each atom is introduced. I assume that the perturbation due to $u(0, 2\pi)$ is random for each atom, so $u(0, 2\pi)$ depends on the site i ? In the present form, $u(0, 2\pi)$ seems to be the same for all sites.

4) I had difficulty understanding the plots Figs. 2f, 2g, and 3b, 3c, 3e. At first, I wondered why we observe that w_{avg} (color) is m dependent. I now guess the authors first plotted w_m vs. m for a specific sample, obtained curves like Figs. 2d and 2e, then calculated w_{avg} , and colored the w_m vs. m curve according to the averaged value. They did it for many configurations to obtain the plots like Figs. 2f, 2g, 3b, 3c, 3e. Do I guess it right? It would be good if the authors explained these plots more in detail.

5) For classification of phases using neural network, the authors cited ref. 30, but in the context of application of CNN in quantum wave systems, T. Ohtsuki et al., J. Phys. Soc. Jpn. 85, 123706 (2016) and P. Broecker, et al. Sci. Rep. 7, 8823 (2017) should also be cited. The former analyzed 2D localized/delocalized wavefunctions via CNN, closely related to the present work. In addition, it might be better to cite recent review articles such as Carleo, et al, Rev. Mod. Phys. 91, 045002, and T. Ohtsuki et al. J. Phys. Soc. Jpn. 89, 022001 (2020).

6) In line 154, the citation 47 has strange commas in front of it.

Reviewer #3 (Remarks to the Author):

The paper describes using ML to analyze properties of disordered materials. The paper correctly notes that this is a challenging problem, and it adopts the approach of converting input data on the disordered system to an image representation in order to use computer vision and NN techniques, thereby identifying several interesting properties. Some of the results are very

interesting--even from the ML side since there is interest in using physics ideas in ML and constructing ML models with physical insight--but there are also aspects that seem confused qua ML.

Below are some more specific comments, in rough order of where they appear in the text, in order to help clarify this.

The title says that ML "imprints" scale-free properties, and the abstract says that ML is "revealing" scale-free properties. It seems to me that neither is the case. Basically, the main ML novelty is a method to identify, in a reasonably data-driven way heavy-tailed or power-law or scale-free properties in disordered materials. That is interesting. The claims about what ML is doing should be more precise/correct.

I am confused by the terminology "wave-to-matter" CNN and "matter-to-wave" CNN in light of the scale-free claims. It seems like two different NNs are constructed, based on the physics, and then they are trained. I think the names have to do with localization of waves and generation of disordered matter, but I'm not sure. Some more details here, both why the names, but more importantly that this is basically a proposal on how to model the data, i.e., a data modeling/representation step.

There is also discussion of scale-free wave networks. I think that is confusing. The proposed method uses a NN, and there has been a lot of work on networks in network science, some of which are cited, but the use of the word network in those two cases is quite different. Here, there are scale-free properties in weight matrices or other properties of a machine learning model, which is not a graph, in the network science sense of the word. Be clear here.

If you are going to relate ML to networks, it would be good to cite more than Refs 1-5, which are the usual network science references, but which most ML people at least hold in relatively low regard (for good reasons, qua ML metrics). I think the technical content of this paper doesn't need to refer itself to those network science papers.

Going beyond traditional approaches that are problem specific to extract microstructural information seems to be the main contribution. That is very interesting. The way this is accomplished is not by using "a data-driven strategy using deep learning" (which, for all I know, is consistent with the vacuous statement that "ML solves the problem"). Instead, the novelty in this paper is in carefully preprocessing data in such a way that NN models (which have proven successful in other areas) can be successfully used to identify some scale-free properties. (By the way, that is essentially the *modus operandi* adopted by state of the art successful NN models in computer vision, natural language processing, etc.)

The fragility/robustness properties of network science networks is somewhat confusing, given that these NN models are not graphs, in the network science sense of the word. There could still be a sensitivity/non-sensitivity, depending on how the input is perturbed, but presumably that has to do with the heavy tailed or extreme value properties more generally. There is a lot of work in the NN area discussing this.

I'm a little surprised how few realizations are needed to get the scale-free behavior. Typically NNs require an enormous amount of data. It would be good to understand better why this is the case, since this is intimately related to why the novel data preprocessing and modeling approach works. Perhaps plots showing how quality varies as a function of data, etc. Diagnostics such as that are important in the NN area. Think of it as a control in a scientific experiment. One of the claims in this paper is that a method is succeeding in a certain way. Run some controls to prove that claim.

Similarly, it would be good to see how the claimed properties don't exhibit themselves, e.g, depending on the amount of disorder in Figure 1. That is, run the other half of a control

experiment.

It is said "To avoid overfitting, different random seeds ..." There is a lot of work now discussing how NNs fit and don't overfit in ways that traditional models do. I suspect that the different random seeds simply adds a bit more data and doesn't have anything to do with overfitting. Prove me right or prove me wrong.

Are Eqn 3 and Eqn 4 the objective functions that are fit to, i.e., that errors are backpropogated from? Be clear.

I think the comments "In the deep learning community, several different approaches have been proposed to resolve this non-uniqueness problem" is not the case, or at least it is not unique to them, which seems to be suggested. Clarify this, since I'm not sure what is being said.

Figure 3 is hard to read, even in the larger version, and a little confusing, and it seems like an important figure. I can't tell, but it may be showing that scale free properties don't hold, for certain parameter values. If so, that is part of the control experiment I mentioned above. If not, then it would be good to see, since it is known that physically this behavior only happens in certain cases. It would be good to see the method fail to reproduce scale-free behavior in two cases: well-trained models, trained to physical data that do not exhibit this; and poorly-trained models, trained to physical data that do exhibit this. Part of the novelty here is a novel use of a powerful ML method. This is like using a powerful new experimental apparatus, and it is important to know what is "real" and what is an artifact of the experimental apparatus.

The scale-invariant behavior in Figure 4 is suggestive, but lines on log-log plots are notoriously hard to read. Given that this is the main claim, I'd suggest a more detailed analysis using the methods of Aaron Clauset et al. Also, Jeff Aslott's powerlaw package (<https://pypi.org/project/powerlaw/>) is one which I have found to be particularly helpful.

More relevant that Barabasi-Albers network science ideas to the scale-free properties discussed here are the heavy tailed behavior in computer vision and other models observed by Martin and Mahoney (<https://arxiv.org/abs/1901.08278>, <https://arxiv.org/abs/1810.01075>) using the weightwatcher (<https://pypi.org/project/WeightWatcher/>) tool. They have a theory to explain why one gets different scale free properties, or not, depending on properties of the model. The current paper wants to say that scale free properties of data are extracted. If I had to guess, this has a lot to do with data preprocessing and the details of the architecture, and less to do with ML per se. That's not a criticism, it's likely a good lesson for this paper. State-of-the-art in computer vision and natural language processing succeeds for the same reason. If that is the case, and if it can be shown (e.g., by running controls on these models, to justify the claims about how/why the ML methods are performing as they do, that that is very interesting. Plus, those papers explicitly discuss when/how "the training process of deep NNs could inherently possess the scale-free property. This physical system is quite different than what they considered. But validating their results in this very different domain, or showing that one gets similar results for a very different reason are both interesting. Doing a detailed analysis using these methods would go a long way to explaining why the method behaves as it does, going well beyond relatively superficial network science analogies.

Reply to Reviewer 1's report

This paper presents a neural network design to learn and convert disorder in a 2d lattice to electron localization modes (waves) and vice-versa. The methodology is mostly sound, but some architectural design choices, as well as choices in training procedure need clarification for me. Also, I am a bit skeptical about their interpretation of the observation about the fat-tailed disorder distribution in the parts where ML doesn't match the theory. If I find their clarifications satisfactory, I would recommend this for

Comments:

The idea on training ML with random functions and asking it to produce target functions is great and should be used more in material design and other branches of physics. I like the idea of the paper, though the scale-free aspect is a strange observation. I am surprised and don't know if I find enough insight in the paper as to why ML chooses these solutions. If the authors manage to convince me they understand why this happened and what is going on I may recommend this for publication. Right now, there are quite a few seemingly arbitrary choices in the architectural design as well as this surprising observation which may well be just an artifact of the architecture. My detailed comments below.

First, we sincerely appreciate the reviewer's positive opinions on our main idea. The reviewer's constructive and valuable comments, especially for the effect of the ML architecture, are significantly helpful to understand the origin of scale-free properties in ML-generated materials. We made every effort to address three main points according to the reviewer's comments:

- I. Examining a different ML architecture (reduced pooling layers) to confirm the generality of scale-free properties**
- II. Examining scale-free properties for a different number of realizations, different degrees of localization, and different test accuracies**
- III. Investigating the relationship between ML architectures and scale-free properties**

Based on this revision, we demonstrated that:

- I. The ML architecture with a reduced number of pooling layers also shows a heavy-tailed distribution with scale invariance (**Supplementary Note S6**).
- II. The scale invariance in ML-generated materials is generally observed in different number of realizations, degrees of localization, and test accuracies (**Supplementary Note S3, Supplementary Note S4, and Fig. 5c,d**).
- III. Scale-free properties are strongly related to the ML architecture as shown in the statistical analysis of the weight distribution (**Fig. 5b and Fig. S9**).

We now convince that the revised manuscript provides concrete evidence about scale-properties in ML-generated materials.

1. My guess is that the scale-free part has to do with the peculiar architectural design chosen by the authors. For example, the 3 levels of max-pooling coarse-grains the disorder by a factor of 8. This exponential decrease in size for each maxpool may allow "fat-tailed" distribution of disorder have similar effects to a normally distributed one. But do the two types of disorder really produce the same physics, or is it just that this ML design is not capable of telling them apart?

We sincerely thank the reviewer for this expert and valuable comment. We agree that the microstructural statistics of ML-generated materials is strongly related to the ML architecture. **Inspired by this comment and Comment 6-2, we examined the different architecture having a reduced number of pooling layers, demonstrating a heavy-tailed distribution and scale invariance in this new architecture.**

In Supplementary Note S6, we compared the original 3-pooling-layer ML (main text) and the newly-designed 1-pooling-layer ML, which include the comparable number of weight and bias parameters for a fair comparison. Figure S7 shows the 1-pooling-layer ML architectures of the matter2wave (which is now changed to “**disorder-to-localization: D2L**”) and wave2matter (which is now changed to “**localization-to-disorder: L2D**”) CNNs.

* The number of ML parameters in each design:

1. The original 3-pooling-layer design
: 14819840 parameters in the D2L CNN and 15342080 parameters in the L2D CNN;
2. The new 1-pooling-layer design
: 17401104 parameters in the D2L CNN and 17925504 parameters in the L2D CNN).

Fig. S7. D2L and L2D CNNs with a single pooling layer. a, b, The network structure of the a, D2L and b, L2D CNN. The details of the network parameters are shown in Supplementary Note S6.

Figure S8 compares the microstructural statistics of the seed and ML-generated structures in the original design (Fig. S8a) and the new design (Fig. S8b). In both cases, the ML-generated class follows power-law statistics (inset (a-1) and (b-1)) and possesses a “heavy-tail” distribution (inset (a-2) and (b-2)). However, the shape and range of the “tail” are dependent on the ML architecture with different power-law fitting results ($\sim \Delta r^{-3.79}$ in the 3-pooling-layer ML and $\sim \Delta r^{-4.09}$ in the 1-pooling-layer ML). **Therefore, this result shows that a heavy-tailed distribution and scale invariance are also observed in a reduced number of pooling layers but the statistical distribution can be controlled according to the ML architecture.**

Fig. S8. Heavy tails in different ML architectures. Statistical distributions of **a**, the 3-pooling-layer design and **b**, the 1-pooling-layer design. Details are shown in Supplementary Note S6.

To guarantee that the observed scale-free properties are not the results of the mismatch between the ML result and theoretical truth, **we also conducted several control experiments: examining scale-free properties for different test accuracies, number of realizations, and degrees of localization.**

First, we examined the effect of the test accuracy (Fig. 5c,d) on the scale invariance. Among 3200 realizations in the example in Fig. 4, we select the sets of ML-generated structures having high ($\geq 84\%$, 194 realizations, Fig. 5c) and low ($\leq 69\%$, 191 realizations, Fig. 5d) test accuracies. We note that both cases possess very similar statistical distributions with the power-law fitting result.

Fig. 5. Relationships between ML-generated disordered structures and test accuracies in terms of scale invariance. **a**, Power-law fitting of the statistical distribution of Δr in ML-generated structures. **c,d**, Power-law fitting results of the realizations for **c**, high ($\geq 84\%$) and **d**, low ($\leq 69\%$) test accuracies.

We also examined the reliability of scale-free properties observed in the main text, by analysing the effect of data size (Supplementary Note S3). The result shows that the unique statistical distribution of ML-generated structures is maintained for a small number of realizations; the power-law exponent α is saturated with $\sim 10^1$ (2560 atoms) realizations, while the lower bound of the heavy tail Δr_{\min} is saturated with $\sim 10^2$ (25600 atoms) realizations. Notably, even a single realization also has a similar value of α and Δr_{\min} .

Fig. S3. Dependence of power-law parameters on the number of realizations. **a**, Power-law exponent α . **b**, The lower bound of the heavy tail Δr_{\min} .

We also note that the scale invariance of ML-generated disordered structures is generally observed for varying degrees of localization (Supplementary Note S4), which strongly implies that the identification of scale-invariant disordered structures originates from the properties of the L2D CNN, not from the mismatch between the ML result and theoretical truth.

2. Also, I am a bit opinionated about the choice of architectural design, as I outline below, but as a first work and proof of concept I think their design is ok. In general I am positive about this paper and think it presents a nice way of using synthetic data with random initial conditions to learn a nonlinear map between two phenomena, in this case disorder and localization wave pattern, and then use the learned mapping to produce a desired wave pattern. I have personal experience with this approach in a different context and it seems to work very well.

We thank the reviewer for his/her encouraging comment and expert advice on our manuscript. We agree that there are many possible choices in the ML architecture to investigate the relationship between localization and disordered materials, and our design is for proof of concept: (i) the ML learning of the relationship between wave localization and disordered materials, and (ii) the identification of disordered patterns with scale-free statistics according to the ML network structure.

As shown in our response to Comment 5 (Fig. 5a,b, and Fig. S9 in Supplementary Note S6), the ML architecture affects the microstructural statistics of ML-generated materials, while maintaining a heavy-tailed distribution and scale-free properties. At this stage, as shown in our response to Comment 1 (and also following the reviewer’s suggestion in Comment 6-2), we focus on the alternative ML architecture that reduces the effect of coarse-grains due to max-pooling (a single pooling ML network shown in Supplementary Note S6).

3. The way the wave-to-matter CNN is trained is also interesting. Usually, one encounters such a design in generative Adversarial networks (GAN). But in GAN, one trains the generator (wave2matter) and the discriminator (matter2wave) at the same time, because otherwise the trained discriminator would be so good that there would be no hope for the generator to converge. So, I am a little surprised that it works so well with a fixed, pre-trained matter2wave. I suspect that is because of the continuous nature of the w_m used in the loss function. Any comments?

We appreciate your insightful and positive feedback on the performance of our CNN design. **We agree with your opinion that the successful convergence of our network is attributed to the continuous nature of w_m in the loss function. Yet we also emphasize that the most critical origin that leads to high accuracy is solving the non-uniqueness problem due to the one-to-many relationship between localization and disordered materials.**

- I. **The use of a pre-trained D2L network:** First of all, the discriminator (now, “disorder-to-localization: D2L”) represents the machine-learning-approximated physical law: the localization calculated from the quantum-mechanical tight-binding equation in Eq. (1). **We thus believe that the test accuracy of the discriminator has to be maximized to guarantee the physical reality of the obtained solution. For this purpose, we employed the pre-trained discriminator with a high test accuracy: ~94.80% test accuracy.**

In this context, we were worried that the competitive (or interactive) learning of the generator (localization-to-disorder (L2D)) and discriminator (disorder-to-localization (D2L)) may guarantee the maximized test accuracy of the “entire” inverse design process but may not guarantee the maximized accuracy for physical reality (the exact tight-binding equation). **However, we fully agree that the application of the reinforcement or unsupervised learning to the design of disordered systems and the following statistical analysis of ML-generated materials in terms of scale-free properties would be an excellent research topic.** We thus included the related discussion in the Discussion section.

(Page 21) “For the inverse design of disordered systems and the following statistical analysis of ML-

generated materials in terms of scale-free properties, the applications of reinforcement learning, unsupervised learning, or well-trained ML networks such as U-net^{59,60} would also be an excellent topic for study.”

II. **The origin of high accuracy:** One of the most critical issues in the ML-based inverse design of physical systems is the non-uniqueness problem. Although there exists a one-to-one relationship between a wavefunction (or optical/acoustic mode) and quantum-mechanical potential (or optical/acoustic material) due to the uniqueness theorem, **wave properties related to the “observable” of a wavefunction (such as localization or spectrum) are non-unique for a given potential.** This issue has been continuously treated in many different fields in physics, for example, the famous question: “Can one hear the shape of a drum?” [50].

This non-uniqueness problem prohibits the convergence in the optimization process [31-35]. For example, there can exist several different configurations of disordered materials D_{p1}, D_{p2}, \dots having the identical localization data W_p . The ML structure may be designed to lead to D_{pq} for W_p according to the ML architecture, and then the loss function will then be defined as $|D_p^{\text{ML}} - D_p^{\text{True}}|$. However, in wave physics, there is no general rule to design the training set $\{D_{p1}, D_{p2}, \dots\}$ or predict the size of this set. It is therefore difficult to train the ML to determine D_{pq} from W_p , because the training set probably includes random configurations of $[D_{11}, D_{23}, D_{37}, D_{42}, \dots]$ for the spectra $[W_1, W_2, W_3, W_4, \dots]$. The complete pre-processing to obtain $[D_{1q}, D_{2q}, D_{3q}, D_{4q}, \dots]$ for the spectra $[W_1, W_2, W_3, W_4, \dots]$ is a very difficult and time-consuming task especially for disordered systems, because many orders/types of order metrics are required to completely represent disordered materials [1].

Our approach defines the loss function $|W_p^{\text{ML}} - W_p^{\text{True}}|$ by connecting the pre-trained CNN to the inverse design CNN. This approach resolves the non-uniqueness problem because we only apply the training set of localization data $[W_1, W_2, W_3, W_4, \dots]$ without the data for disordered materials. The resulting microstructural information D_{pq} for W_p is then determined by our autoencoder-type ML architecture. **Notably, the control of the statistical distribution of disordered materials using a different ML architecture (Fig. S8) demonstrates the change from $[D_{p1}]$ to $[D_{p2}]$ for the same localization data $[W_p]$.** We included the discussion to clarify this issue.

(Page 9) “This one-to-many relationship between a wave property and matter has made it difficult to achieve a stable inverse design of material from a given wave property because the existence of many solutions (matter) for an input (wave property) prohibits the stable convergence of the optimization for a cost function.”

4. Are "accuracies" test accuracies? It should be made clear that they are not simply training accuracy.

We thank the reviewer for his/her careful reading of our manuscript. We revised the manuscript clarifying the use of the training cost function, validation accuracy, and test accuracy.

5. Fig 3. a) what is the intuition behind the design? At first sight, you are building an auto-encoder for the localization. But you also want it to be a decoder which converts localization to disorder, hence the reshaping, etc. What I don't understand is why you first make the localization wave spatial, then flatten it to a fully connected network and then reshape again to get the disorder image. I guess you want the FC layers to learn the non-linear wave-to-matter transformations, right?

As we stated in our response to Comment 3, we used the autoencoder-type ML architecture, which represents the inverse-design encoder (L2D CNN) and physical-law decoder (D2L CNN). For physical reality, we started from the D2L CNN with the maximum test accuracy.

For the use of FC layers, the reviewer's comment is accurate. We tried to learn the nonlinear wave-to-matter relationships through the FC layer because the FC layer has been traditionally used to map the extracted (and downsampled) features from the CNN into output. **In this revision, we utilized this relationship between the FC layer and output neurons to understand the effect of ML architectures on ML-generated materials.**

For w_{ji}^x and w_{ji}^y , which denote the weights from the i^{th} FC neuron to the j^{th} x -axis and y -axis output neurons, respectively, we defined the strength of the weights to the j^{th} output neuron (or the j^{th} atom in an ML-generated material) as $W_j = \sum_i [(w_{ji}^x)^2 + (w_{ji}^y)^2]$. Figure S9a,b represents the comparison of the relationships between ML architectures and ML-generated structures in the 3-pooling-layer. Figure S9b shows a very similar statistical distribution with Fig. S9a and also possesses the heavy-tailed distribution with scale-invariance. **The comparison between Figs. S9a and S9b provides clear-cut evidence of the effect of the ML architecture on ML-generated materials.**

This finding becomes more evident by examining different ML architectures. In Fig. S9c,d, we investigated the ML architecture with a single pooling stage. The heavy-tailed distribution is also maintained in this design. Furthermore, the decrease of the "tail length" in the 1-pooling-layer design (Fig. S9c versus Fig. S9a) originates from the decreased range of the weight strength W_j (Fig. S9d versus Fig. S9b), which shows that **the control of the ML architecture leads to the corresponding alteration of ML-generated structures.**

Fig. S9. Relationships between disordered materials and ML networks for different ML architectures. a, b, 3-pooling-layer ML design and c, d, 1-pooling-layer ML design.

6. Architecture, personal opinion:

Matter-to-Wave CNN:

1. Both the mode area array and the disorder have size = N. In the Matter-to-Wave CNN, why do you not try to convert w_m to 16x16 and use convolutional output layers, as you would do with an auto-encoder or the U-net architecture? U-Net, in particular has been observed to do very well with converting two functions over a lattice to each other.

Thank you very much for your insightful comment. In the Matter-to-Wave (now, D2L) CNN, the localization data w_m is converted to the 1D array to represent the modal localization. In the Wave-to-Matter (now, L2D) CNN, the data w_m is converted to the 16×16 data, and the data for materials are converted to the 1D array to exploit the FC layers that enable to find nonlinear wave-to-matter relationships through the mapping of the extracted features from the CNN into output (Figs. 5 and S9). **However, we agree that the U-net architecture will be beneficial to improve the device performance. We included the related discussion and references [59,60] in the Discussion section.**

2. You have a 16x16 input. After 3 layers with maxpooling, you have at most a $2 \times 2 \times (\text{num filters})$ output, so most spatial info is extremely coarse-grained. Then, instead of a traditional upsampling CNN and 16x16 w_m output, you are using FC layers to get a vector w_m . Can you motivate this? Did you try any other architecture for your decoder?

Following the reviewer's advice, we tested the ML architecture having a reduced number of pooling layers. The result is shown in our response to Comments 1 and 5 (Supplementary Note S6). We sincerely thank the reviewer's very kind and expert comments in this peer review.

Reply to Reviewer 2's report

The authors have shown that convolutional neural network (CNN) can predict the inverse participation ratio (IPR) of wavefunctions from the distribution of atoms (Matter-Wave CNN), as well as the distribution of atoms from the IPR (Wave-Matter CNN). The obtained distributions of atoms show scale-free properties, which is demonstrated to be robust against perturbations, namely, the fluctuations of positions of atoms. The results are interesting, but already more than three years have passed since the numerous machine learning techniques was applied to condensed matter physics. I therefore have reservation to recommend publication of this article in Nature Communications since the impact of this research is no longer matches the high publication criterion. The paper may be more suitable in publication in Scientific Reports, but before I recommend publication, I have several comments/questions that might improve the paper and its readability.

We sincerely thank the reviewer for his/her thoughtful and professional review of our manuscript. Following the reviewer's comments, the quality of our manuscript was significantly improved.

Meanwhile, we cordially hope the reviewer approves of the significant improvements made in this revision. **We convince that the main results of our manuscript,**

- **the importance of the ML architecture on material design, and**
- **the finding of scale-free properties in ML-generated structures,**

have not been reported despite numerous researches conducted in this field during the last few years. Most of the researches have focused on achieving more accurate prediction and design of wave-matter interactions. The relationship between ML architecture and the resulting physical phenomena especially in terms of heavy-tailed distributions and scale-free properties has been neglected.

To clarify this novelty, in this revision, we made every effort to demonstrate:

- I. The connection between scale-free properties and ML architectures**
- II. The generality of scale-free properties for data size, localization, and test accuracies**
- III. The control of scale-free properties with different ML architectures**

The newly included contents demonstrate that:

- I. Scale-free properties originate from the ML network parameters as shown in the matched statistical distribution between material parameters and network weights (**Fig. 5a,b, Fig. S9**),
- II. The scale-free properties are universally maintained for varying number of realizations, degrees of localization, or test accuracies (**Supplementary Note S3, S4, and Fig. 5c,d**), and
- III. The change of the ML architecture leads to the manipulation of scale-free properties (**Supplementary Note S6**).

From these findings, we demonstrated the method of generating materials, which inherit the advantages of scale-free systems: defect-robustness (robust to random attacks) and high modulation efficiency (sensitive to target attacks). These achievements will provide novel design freedom for active devices in condensed-matter physics, photonics, and acoustics. With the in-depth revision including the generalization of our study, together with the positive responses to our original findings from the reviewers, we cordially request your reconsideration of our manuscript.

1) In equation (2), w_m is written that it is defined by the IPR. This sentence is a bit confusing, since w_m is the inverse of IPR, not IPR itself. The authors should write explicitly that w_m is the inverse of IPR.

We are sorry for the mistake and thank the reviewer very much for his/her careful reading of our manuscript. We revised the phrase to clarify that w_m is the inverse of the IPR.

(Page 5) "... w_m , which is defined by the inverse of the inverse participation ratio (IPR) as"

2) In Fig. 2, the on-site energy epsilon is set to 1. Do we need this? The energy epsilon is just the shift of whole energy spectrum, and the localization properties should be independent of epsilon.

The reviewer is right. The on-site energy, which is identical to all of the atoms, does not affect mode areas (or localization). However, we set the on-site energy in Fig. 2 because all of the lattice information is introduced in Fig. 2 in the main text and we also observe energy spectra in the later discussion (Supplementary Note S5). We clarified this point in the revision.

(Fig. 2 Caption) "Although the on-site energy does not affect mode areas, we set the on-site energy as $\epsilon = 1$ for energy spectra in the later discussion (Supplementary Note S5)."

3) In the section of "scale-free wave networks with hub atoms," the position perturbation of each atom is introduced. I assume that the perturbation due to $u(0, 2\pi)$ is random for each atom, so $u(0, 2\pi)$ depends on the site i ? In the present form, $u(0, 2\pi)$ seems to be the same for all sites.

We are sorry for the confusion. Although we tried to represent that $u(a, b)$ generates a random value at each atom, we revised the expression as $u_i(a, b)$ for more clarity following the reviewer's suggestion. We also revised Methods and Supplementary Note S1 according to this change.

(Page 18) "... $u_i(a, b)$ is the random value for the i^{th} atom from the uniform random distribution ..."

4) I had difficulty understanding the plots Figs. 2f, 2g, and 3b, 3c, 3e. At first, I wondered why we observe that w_{avg} (color) is m dependent. I now guess the authors first plotted w_m vs. m for a specific sample, obtained curves like Figs. 2d and 2e, then calculated w_{avg} , and colored the w_m vs. m curve according to the averaged value. They did it for many configurations to obtain the plots like

Figs. 2f, 2g, 3b, 3c, 3e. Do I guess it right? It would be good if the authors explained these plots more in detail.

The reviewer's statement is accurate. We also sincerely appreciate the reviewer's excellent description of Figs. 2f, 2g, and 3b, 3c, and 3e. We included more detailed discussion on these figures following the reviewer's description.

(Pages 7-8) "We also compare the ground-truth (Fig. 2f) and ML-predicted (Fig. 2g) localization for a wide range of localization values of the test dataset (1×10^4 realizations). Figures 2f and 2g are obtained by plotting w_m of each realization as a function of the mode number m and colouring each point according to the average mode area $w_{\text{avg}} = \sum w_m / N$ of each realization. We note that the ground-truth and ML-predicted localization shows excellent agreement for different values of w_{avg} , achieving the test accuracy $1 - L_{\text{D2L}} \sim 94.80\%$."

(Page 10) "We compare the target localizations (Fig. 3c) to the ML-predicted localizations obtained through the L2D2L CNN (Fig. 3d) and the Hamiltonian-calculated true values of the disordered structures generated by the L2D CNN (Fig. 3e), using the same data plotting format with those in Fig. 2f,g."

5) For classification of phases using neural network, the authors cited ref. 30, but in the context of application of CNN in quantum wave systems, T. Ohtsuki et al., J. Phys. Soc. Jpn. 85, 123706 (2016) and P. Broecker, et al. Sci. Rep. 7, 8823 (2017) should also be cited. The former analyzed 2D localized/delocalized wavefunctions via CNN, closely related to the present work. In addition, it might be better to cite recent review articles such as Carleo, et al, Rev. Mod. Phys. 91, 045002, and T. Ohtsuki et al. J. Phys. Soc. Jpn. 89, 022001 (2020).

We appreciate the reviewer for suggesting a proper list of related references. We included these references as [24,25,29,30] with additional discussion. The review articles references [24,25] provide a general overview of recent applications of deep learning in physics problems. The reference [29] is also an important paper in deep learning for fermionic quantum systems. The reference [30] also shows an interesting result on the inference of the phase of matter using wavefunctions in disordered systems.

Meanwhile, we also emphasize that our main result with the inference of microstructures from quantum-mechanical observables (localization) shows a novelty when compared with those remarkable achievements, in terms of (i) the identification of disordered materials having scale-free properties and (ii) the clarification of the effect of ML networks on material design. We made every effort to clarify the novelty and impact of our manuscript by following all of the valuable comments and suggestions raised by the reviewers thoroughly.

(Page 3) "Because of its applicability to general-purpose data formats, deep learning has recently been extended to handle a number of physics problems^{24,25}, such as classifications of crystals²⁶ or topological order²⁷, phase transitions and order parameters²⁸⁻³⁰, optical device designs³¹⁻³⁶, and image reconstructions³⁷. When we consider the vast amount of design freedom in disordered systems, deep learning will compose a powerful

toolkit for resolving complexities in wave behaviours inside disordered structures, as shown in the inference of phases of matter using eigenfunctions^{25,30}.

6) In line 154, the citation 47 has strange commas in front of it.

Thank you for your careful suggestion. We revised that part to avoid the confusion.

(Page 9) (Previous) "... a classic question of "Can one hear the shape of a drum?"⁴⁷ and ..."

→ (Revised) "...a classic question⁵⁰ of "Can one hear the shape of a drum?" and ..."

Reply to Reviewer 3's report

The paper describes using ML to analyze properties of disordered materials. The paper correctly notes that this is a challenging problem, and it adopts the approach of converting input data on the disordered system to an image representation in order to use computer vision and NN techniques, thereby identifying several interesting properties. Some of the results are very interesting--even from the ML side since there is interest in using physics ideas in ML and constructing ML models with physical insight--but there are also aspects that seem confused qua ML.

We sincerely appreciate the reviewer's positive opinions on our results. The reviewer's constructive and expert comments are undoubtedly helpful for improving our manuscript, especially in terms of the rigorous analysis of scale-free properties and their relationship to ML architectures. During this revision, we made every effort to address three main points:

- I. Employing precise concepts and analysing tools on scale-free properties and ML**
- II. Analysing scale-free properties in different number of realizations, ML architectures, test accuracies, and degrees of localization**
- III. Investigating the relationship between ML weights and scale-free properties in materials**

Based on this revision, we demonstrated that:

- I. Scale-free properties in ML-generated materials are consistent under statistically rigorous methods suggested by Aaron Clauset *et al.*,
- II. Scale-free properties are generally observed in various control experiments: a different number of realizations (**Supplementary Note S3**), a different ML architecture (with a reduced number of pooling layers, **Supplementary Note S6**), different test accuracies (**Fig. 5c,d**), and different degrees of localization (**Supplementary Note S4**), and
- III. Scale-free properties are strongly related to the ML architecture (more exactly, the weight distribution) as the reviewer predicted (**Fig. 5b and Fig. S9**).

We convince that the revised manuscript now provides concrete evidence about scale-properties in ML-generated materials.

Below are some more specific comments, in rough order of where they appear in the text, in order to help clarify this.

The title says that ML "imprints" scale-free properties, and the abstract says that ML is "revealing" scale-free properties. It seems to me that neither is the case. Basically, the main ML novelty is a method to identify, in a reasonably data-driven way heavy-tailed or power-law or scale-free properties in disordered materials. That is interesting. The claims about what ML is doing should be more precise/correct.

Thank you very much for the accurate description and insightful comment of our results. **Because disordered materials with scale-free properties already exist whether or not the ML is applied, we fully agree with the reviewer’s opinion that our work is to “identify” a set of disordered materials that have scale-free properties.** The role of the ML is then to identify or distinguish such scale-free-type materials using given localization data. We carefully revised our manuscript following the reviewer’s description.

(Title) “Machine learning identifies scale-free properties in disordered materials”

(Abstract) “Here, we employ a machine learning (ML) approach for predicting and designing wave-matter interactions in disordered structures, thereby identifying scale-free properties for waves.”

I am confused by the terminology "wave-to-matter" CNN and "matter-to-wave" CNN in light of the scale-free claims. It seems like two different NNs are constructed, based on the physics, and then they are trained. I think the names have to do with localization of waves and generation of disordered matter, but I'm not sure. Some more details here, both why the names, but more importantly that this is basically a proposal on how to model the data, i.e., a data modeling/representation step.

We agree with your suggestions that more accurate terminologies and in-depth descriptions of the principles of data modelling are necessary. First, we changed the terminologies “wave-to-matter” CNN and “matter-to-wave” CNN to “localization-to-disorder (L2D)” CNN and “disorder-to-localization (D2L)” CNN. We also included more detailed descriptions of (i) the role of each CNN and (ii) the training processes of these CNNs.

(Abstract) “..., we develop disorder-to-localization and localization-to-disorder convolutional neural networks (CNNs).”

(Page 9) “To guarantee the physical reality of the obtained solution, we utilize the “trained” D2L CNN with the fixed weight and bias parameters...”

(Page 10) “We note that although the training dataset for the L2D2L CNN consists of localization data obtained from the tight-binding Hamiltonian in Eq. (1), the microstructural information used for the target localization data is not applied to the training of the L2D2L CNN.”

There is also discussion of scale-free wave networks. I think that is confusing. The proposed method uses a NN, and there has been a lot of work on networks in network science, some of which are cited, but the use of the word network in those two cases is quite different. Here, there are scale-free properties in weight matrices or other properties of a machine learning model, which is not a graph, in the network science sense of the word. Be clear here.

We thank the reviewer for his/her thoughtful advice on the exact use of concepts and terminologies. We agree that the concepts of the graph “network” in network science and the neural “network” in machine learning were confusingly discussed in our previous manuscript. We thoroughly revised the entire manuscript following the reviewer’s suggestion, focusing only on the neural “network” in machine

learning which is the main interest of our work. All the other meaning of the network (“or graph”) was removed.

If you are going to relate ML to networks, it would be good to cite more than Refs 1-5, which are the usual network science references, but which most ML people at least hold in relatively low regard (for good reasons, qua ML metrics). I think the technical content of this paper doesn't need to refer itself to those network science papers.

We agree with the reviewer’s expert comment. We acknowledge that the connection of our work to network science based on graph networks may raise confusion. We revised some confusing discussion for clarity. We also removed previous ref. 2 for small-world networks, which is not related to our work.

Going beyond traditional approaches that are problem specific to extract microstructural information seems to be the main contribution. That is very interesting. The way this is accomplished is not by using "a data-driven strategy using deep learning" (which, for all I know, is consistent with the vacuous statement that "ML solves the problem"). Instead, the novelty in this paper is in carefully preprocessing data in such a way that NN models (which have proven successful in other areas) can be successfully used to identify some scale-free properties. (By the way, that is essentially the modus operandi adopted by state of the art successful NN models in computer vision, natural language processing, etc.)

Thank you very much for this valuable advice. The reviewer’s description on the novelty and methodology of our result is accurate. We carefully revised our manuscript thoroughly to reflect the reviewer’s accurate description.

(Introduction) “To substitute the time-consuming and problem-specific process of calculating microstructural descriptors while making full use of microstructural information, we can envisage the use of multiple-layer neural network (NN) models to identify the relationship between disordered structures and wave behaviours. This deep-learning-based framework^{20,21}, one of the powerful machine learning (ML) tools, has proven successful for abstracting the features of datasets in pattern recognition, decision making, and language translation^{22,23} when carefully preprocessed data can be used.”

The fragility/robustness properties of network science networks is somewhat confusing, given that these NN models are not graphs, in the network science sense of the word. There could still be a sensitivity/non-sensitivity, depending on how the input is perturbed, but presumably that has to do with the heavy tailed or extreme value properties more generally. There is a lot of work in the NN area discussing this.

We are sorry for this confusion, and also thank for the reviewer’s thoughtful comment. We revised the manuscript to use the term “network” as a ML network, not a graph network. Because the terminologies of robustness or fragility, which are used for the properties of ML-generated disordered materials in this manuscript, have also been used in material science to represent sensitivity or non-sensitivity, we carefully maintain these terminologies but with sufficient explanations.

I'm a little surprised how few realizations are needed to get the scale-free behavior. Typically NNs require an enormous amount of data. It would be good to understand better why this is the case, since this is intimately related to why the novel data preprocessing and modeling approach works. Perhaps plots showing how quality varies as a function of data, etc. Diagnostics such as that are important in the NN area. Think of it as a control in a scientific experiment. One of the claims in this paper is that a method is succeeding in a certain way. Run some controls to prove that claim.

Similarly, it would be good to see how the claimed properties don't exhibit themselves, e.g, depending on the amount of disorder in Figure 1. That is, run the other half of a control experiment.

We appreciate the reviewer's expert comments on the necessity of various control experiments for our result. We recognized that we need to include a more in-depth discussion and evidence on "why the ML-generated disordered materials possess scale-free properties" and "whether these scale-free properties are numerical artefacts" through control experiments. During this revision, we made every effort to provide answers to these questions.

First, we tested the reliability of the scale-free behaviour by examining the dependence of power-law parameters (power-law exponent α and the lower bound of the heavy tail Δr_{\min}) in **Supplementary Note S3**. The result shows that the unique statistical distribution of ML-generated structures is maintained for a small number of realizations, from $\sim 10^1$ (2560 atoms) to $\sim 10^2$ (25600 atoms) realizations, and even a single realization also provides a similar value of α and Δr_{\min} .

Fig. S3. Dependence of power-law parameters on the number of realizations. a, Power-law exponent α . **b,** The lower bound of the heavy tail Δr_{\min} .

For more complete answers to other control experiments, the related discussion is shown below: the discussion in the reply letter pages 18-20 (I. Different degrees of localization and II. Different test accuracies) and pages 21-23 (Different ML architecture).

It is said "To avoid overfitting, different random seeds ..." There is a lot of work now discussing how NNs fit and don't overfit in ways that traditional models do. I suspect that the different random seeds simply adds a bit more data and doesn't have anything to do with overfitting. Prove me right or prove me wrong.

We are sorry for this evident mistake. The commented sentence was intended to represent the conventional use of the classification of training, validation, and test data to “check” overfitting, not to “avoid” overfitting. As the reviewer accurately pointed out, different random seeds for each data set do not have anything to do with overfitting. We revised this sentence correctly.

(Page 7) “To monitor overfitting during and after the training, different random seeds for the deformation have been used in the training, validation, and test datasets.”

Are Eqn 3 and Eqn 4 the objective functions that are fit to, i.e., that errors are backpropagated from? Be clear.

We revised the related parts to clarify that our objective functions are the mean absolute percentage error (MAPE), which has been widely used in forecasting models with regression and machine learning methods including error backpropagations.

(Pages 6,7) “Because each mode has different degrees of localization, it is necessary to fairly estimate the regression error for a wide range of w_m values. We thus employ the mean absolute percentage error (MAPE) as the cost function, which has been widely applied to regression and machine learning for forecasting models^{47,48}. The MAPE cost function for the D2L CNN is expressed as ...”

I think the comments "In the deep learning community, several different approaches have been proposed to resolve this non-uniqueness problem" is not the case, or at least it is not unique to them, which seems to be suggested. Clarify this, since I'm not sure what is being said.

We are sorry for this inaccurate and somewhat overclaimed sentence. We revised the sentence more correctly with additional explanation.

(Page 9) “This one-to-many relationship between a wave property and matter has made it difficult to achieve a stable inverse design of material from a given wave property because the existence of many solutions (matter) for an input (wave property) prohibits the stable convergence of the optimization for a cost function. In the inverse design of material using the ML method, several different approaches have been proposed to resolve this non-uniqueness problem: ...”

Figure 3 is hard to read, even in the larger version, and a little confusing, and it seems like an important figure. I can't tell, but it may be showing that scale-free properties don't hold, for certain parameter values. If so, that is part of the control experiment I mentioned above. If not, then it would be good to see, since it is known that physically this behavior only happens in certain cases. It would be good to see the method fail to reproduce scale-free behavior in two cases: well-trained models, trained to physical data that do not exhibit this; and poorly-trained models, trained to physical data that do exhibit this. Part of the novelty here is a novel use of a powerful ML method. This is like using a powerful new experimental apparatus, and it is important to know what is "real" and what is an artifact of the experimental apparatus.

We thank the reviewer’s valuable comment. First, for the clarity of the CNN structure in Fig. 3, we included Fig. 3a to clearly present the structure of the L2D CNN in detail. More detailed discussion was also added to explain the plotting format of Fig. 3c-e. Especially, we completely agree that we need additional control experiments, to explicitly examine the generality of scale-free properties and therefore demonstrate that scale-free properties are consistently observed in our design. In our previous manuscript, we examined scale-free properties only for restricted conditions: restricted ranges of localization ($0.20 \leq w_{\text{avg}} \leq 0.30$) and the entire 3200 realizations which include the realizations of varying test accuracies. To extend this observation, we looked into scale-free properties in:

- I. **Different degrees of localization, and**
- II. **Different test accuracies (high accuracy *versus* poor accuracy)**

I. Different degrees of localization (Supplementary Note S4)

In **Supplementary Note S4** in the revised manuscript, we examined the dependence of scale-free properties in terms of the degrees of localization: the ranges of (i) $0.20 \leq w_{\text{avg}} \leq 0.25$, (ii) $0.25 \leq w_{\text{avg}} \leq 0.30$, (iii) $0.30 \leq w_{\text{avg}} \leq 0.35$, and (iv) $0.35 \leq w_{\text{avg}} \leq 0.40$. As shown in **Fig. S4**, in all the ranges of localization strength (which is directly related to the degree of disorder in materials), we can observe the heavy-tailed distributions in ML-generated structures.

Following the reviewer’s suggestion, we also applied the power-law fitting method demonstrated by Aaron Clauset *et al.* to examine the scale-free properties in different degrees of localization. As shown in **Fig. S5**, we showed that the power-law fitting result is obtained consistently for every observed ranges of localization. Therefore, the scale invariance of ML-generated disordered structures is maintained for varying degrees of localization, although the position of heavy tails changes (black dashed lines in Fig. S5) to meet the target localization condition.

II. Different test accuracies (Fig. 5c,d)

We also tried to guarantee that the observed scale-free properties are not the results of the mismatch between the ML result and theoretical truth. We therefore examined the effect of the test accuracy (**Fig. 5c,d in the main text**) on the scale invariance. Among 3200 realizations in the example in Fig. 4, we separated the sets of ML-generated structures having high ($\geq 84\%$, 194 realizations, **Fig. 5c**) and low ($\leq 69\%$, 191 realizations, **Fig. 5d**) test accuracies. We note that both cases possess very similar statistical distributions with the power-law fitting result, demonstrating that scale-free properties originate from the ML architecture (as demonstrated in our final response in the reply letter pages 21-23, **Supplementary Note S6**).

Fig. S4. Statistical distributions for different degrees of localization. The strength of the lattice deformation Δr is shown for the seed (green) and ML-generated (orange) structures. **a**, $0.20 \leq w_{\text{avg}} \leq 0.25$ (1823 realizations), **b**, $0.25 \leq w_{\text{avg}} \leq 0.30$ (1377 realizations), **c**, $0.30 \leq w_{\text{avg}} \leq 0.35$ (1588 realizations), and **d**, $0.35 \leq w_{\text{avg}} \leq 0.40$ (2123 realizations) in the ML design. Each inset (a-1 to d-1) shows the extended plot near a heavy tail.

Fig. S5. Power-law analysis near heavy tails. Each plot shows the log-log plot of each case in Fig. S4 for the ML design, illustrating the power-law distribution. The orange line (composed of discretized points) represents the complementary cumulative distribution function (CDF) obtained from the data set in Fig. S4. **a**, $0.20 \leq w_{\text{avg}} \leq 0.25$ (1823 realizations), **b**, $0.25 \leq w_{\text{avg}} \leq 0.30$ (1377 realizations), **c**, $0.30 \leq w_{\text{avg}} \leq 0.35$ (1588 realizations), and **d**, $0.35 \leq w_{\text{avg}} \leq 0.40$ (2123 realizations) in the ML design.

Fig. 5. Relationships between ML-generated disordered structures and test accuracies in terms of scale invariance. **a**, Power-law fitting of the statistical distribution of Δr in ML-generated structures. **c,d**, Power-law fitting results of the realizations for **c**, high ($\geq 84\%$) and **d**, low ($\geq 69\%$) test accuracies.

The scale-invariant behavior in Figure 4 is suggestive, but lines on log-log plots are notoriously hard to read. Given that this is the main claim, I'd suggest a more detailed analysis using the methods of Aaron Clauset et al. Also, Jeff Aslott's powerlaw package (<https://pypi.org/project/powerlaw/>) is one which I have found to be particularly helpful.

We sincerely appreciate this expert advice on the power-law analysis of the heavy-tailed distribution. We acknowledge that our previous method was not sufficient for statistically rigorous analysis. **Following your advice, we employed the method proposed and demonstrated by Aaron Clauset et al., which utilizes maximum-likelihood fitting methods with goodness-of-fit tests [52]. The result is also cross-checked with Jeff Aslott's power-law package [53].**

We agree that the applied method, which has been rigorously confirmed in the community, provides a statistically more reliable fitting for our data. In this revision, we changed Fig. 4g-1 and included related references [52,53]. This method is also applied to all the other power-law fitting results shown in Fig. 5 and Figs. S3,S5,S8,S9.

More relevant that Barabasi-Albers network science ideas to the scale-free properties discussed here are the heavy tailed behavior in computer vision and other models observed by Martin and Mahoney (<https://arxiv.org/abs/1901.08278>, <https://arxiv.org/abs/1810.01075>) using the weightwatcher (<https://pypi.org/project/WeightWatcher/>) tool. They have a theory to explain why one gets different scale free properties, or not, depending on properties of the model. The current paper wants to say that scale free properties of data are extracted. If I had to guess, this has a lot to do with data preprocessing and the details of the architecture, and less to do with ML per se. That's not a criticism, it's likely a good lesson for this paper. State-of-the-art in computer vision and natural language processing succeeds for the same reason. If that is the case, and if it can be shown (e.g., by running controls on these models, to justify the claims about how/why the ML methods are performing as they do, that that is very interesting. Plus, those papers explicitly discuss when/how "the training process of deep NNs could inherently possess the scale-free property. This physical system is quite different than what they considered. But validating their results in this very different domain, or showing that one gets similar results for a very different reason are both interesting. Doing a detailed analysis using these methods would go a long way to explaining why the method behaves as it does, going well beyond relatively superficial network science analogies.

We thank the reviewer for his/her expert advice that provides excellent guidance for understanding scale-free properties in ML-generated materials and their relationships to ML networks. Although we had expected that scale-free properties might originate from the ML architecture, **we can examine this expectation owing to the reviewer's suggestion on monitoring the weight parameters in the ML network. The result clarifies that scale-free properties in ML-generated materials originate from the structure of the ML network.**

In our ML network, the lattice deformation of each atom is determined by the value of each pair of x -axis and y -axis output neurons in the L2D CNN, as shown in the revised Fig. 3a. Therefore, among numerous weight and bias parameters (14819840 parameters in the D2L CNN and 15342080 parameters in the L2D CNN), **the most critical parameters are the weights from the FC layer to the output layer in the L2D CNN, which are described by a 2048×512 matrix.** Although the weights and bias in hidden layers should also affect the output layer neurons indirectly, it is difficult to estimate such indirect effects. Also, we expect that these indirect effects are less significant than the direct effect from the FC-output weights.

For w_{ji}^x and w_{ji}^y , which denote the weights from the i^{th} FC neuron to the j^{th} x -axis and y -axis output neurons, respectively ($1 \leq i \leq 2048$ and $1 \leq j \leq 256$ in our design), we define the strength of the weights to the j^{th} output neuron (or the j^{th} atom in a ML-generated disordered material) as $W_j = \sum_i [(w_{ji}^x)^2 + (w_{ji}^y)^2]$. Using the method proposed and demonstrated by Aaron Clauset *et al.*, we analyse the statistical distribution of W_j and compare it with that of ML-generated disordered materials quantified by Δr .

Figure 5b in the main text (or Fig. S9b in Supplementary Note S6) shows the CDF of W_j , which represents a very similar statistical distribution with Δr in terms of its inflection point (Fig. 5a or Fig. S9a) and also possesses the heavy-tailed distribution. **The comparison between Figs. 5a and 5b provides clear-cut evidence of the effect of the ML network structure on ML-generated materials. Because this finding becomes evident especially when examining different ML architectures—controlling the ML model—which lead to different weights and bias distributions, in Supplementary Note S6, we investigated another D2L and L2D CNNs each with a single pooling stage (Fig. S7). The result shows that we can control the W_j distribution by manipulating the ML architecture while preserving the heavy-tailed distribution (Fig. S9d versus Fig. S9b), and this control leads to the following alteration of ML-generated structures (Fig. S9c versus Fig. S9a).**

We also appreciate the reviewer’s excellent suggestion on referring previous works conducted by C. H. Martin and M. W. Mahoney [55,56]. **These works, which demonstrated that the weight matrices of well-trained deep NNs can be fit to a power law with the heavy-tailed distribution, provide an optimal viewpoint on interpreting our result.** We included these references and related discussion.

(Page 21) “Recently, in random matrix theory, it was demonstrated that the correlations in the weight matrices of well-trained deep NNs can be fit to a power law with the heavy-tailed distribution^{55,56}. This theory enables the successful analogy between ML network structures and ML-generated real-space wave structures in our result: the identification of the “heavy-tailed perturbation distribution” of atomic sites using the “heavy-tailed weight distribution” of CNN neurons.”

Fig. S9. Relationships between disordered materials and ML networks for different ML architectures. a,b, 3-pooling-layer ML design and c,d, 1-pooling-layer ML design.